



# Measurement report: The influence of traffic and new particle formation on the size distribution of 1-800 nm particles in Helsinki: a street canyon and an urban background station comparison

Magdalena Okuljar[1], Heino Kuuluvainen[2], Jenni Kontkanen[1], Olga Garmash[1], Miska Olin[2], Jarkko V. Niemi[3], Hilkka Timonen[4], Juha Kangasluoma[1], Yee Jun Tham[1], Rima Baalbaki[1], Mikko Sipilä[1], Laura Salo[2], Henna Lintusaari[2], Harri Portin[3], Kimmo Teinilä[4], Minna Aurela[4], Miikka Dal Maso[2], Topi Rönkkö[2], Tuukka Petäjä[1] and Pauli Paasonen[1]

[1]Institute of Atmospheric and Earth System Science / Physics, Faculty of Science, University of Helsinki, FI-00014, Helsinki, Finland
[2]Aerosol Physics Laboratory, Physics Unit, Tampere University, PO Box 692, FI-33014, Tampere, Finland
[3]Helsinki Region Environmental Services Authority, PO Box 100, FI-00066, Helsinki, Finland
[4]Atmospheric Composition Research, Finnish Meteorological Institute, PO Box 503, FI-00101, Helsinki, Finland

*Correspondence to*: Magdalena Okuljar (magdalena.okuljar@helsinki.fi)

**Abstract.** Most of the anthropogenic air pollution sources are located in urban environments. The contribution of these sources to the population of atmospheric particles in the urban environment is poorly known. In this study, we investigated the aerosol particle number concentrations in a diameter range from 1 to 800 nm at a street canyon site and at a background station within 1 km from each other in Helsinki, Finland. We use these number size distribution data together with complementary trace gas data and develop a method to estimate the relative contributions of traffic and atmospheric new particle formation (NPF) to the concentrations of sub-3 nm particles. During the daytime, the particle concentrations were higher at the street canyon site than at the background station in all analyzed modes: sub-3 nm particles, nucleation mode (3-25 nm), Aitken mode (25-100 nm), and accumulation mode (100-800 nm). The population of sub-3 nm and nucleation mode particles was linked to local sources such as traffic, while the accumulation mode particles were more related to non-local sources. Aitken mode particles were dominated by local sources at the street canyon site while at the background station they were mainly influenced by non-local sources. The results of this study support earlier research showing direct emissions of the sub-3 nm particles from traffic. However, by using our new method, we show that during NPF events, traffic contribution to the total sub-3 nm particle concentration can be small and during daytime (6:00-20:00) in spring it does not dominate the sub-3 nm particle population at either of the researched sites. In the future, this method can be applied to estimate the contribution of traffic to particle number concentrations in different urban environments. This knowledge is needed to evaluate the effects of traffic on urban air quality.

## 1. Introduction

Aerosol particles are both directly emitted to the atmosphere (primary particles) and formed from gaseous precursors (secondary particles) (Kulmala and Kerminen, 2008). Secondary particles can form by new particle formation (NPF) via atmospheric photochemical reactions or nucleate in plumes from local sources (Kerminen et al., 2018; Mylläri et al., 2016). Additionally, particles may form while hot vehicle exhaust is cooled and diluted, which is called delayed primary particulate matter formation (Rönkkö and Timonen, 2019). The urban environment contains a mixture of secondary particles, and primary particles emitted from a variety of industrial processes, traffic, power generation, and natural sources (Seinfeld and Pandis, 2016). Aerosol particles influence the visibility (Hyslop, 2009), the hydrological cycle (Rosenfeld et al., 2008), and radiation balance (Andreae, 2009; Ramanathan and Feng, 2009). Furthermore, particles can



harm human health, impacting respiratory and cardiovascular systems (André, 2014). Urban air pollution also contains
magnetite nanoparticles, which accumulate in the brain, and may cause neurodegenerative diseases (Maher et al., 2016).
Nanoparticles with diameters below 3 nm may have significant, so far poorly understood, health effects due to their nose-
to-brain transport via the olfactory pathway (Tian et al., 2019). Models show that outdoor air pollution causes
approximately 400 000 premature deaths in Europe annually (Geels et al., 2015; Im et al., 2018), from which around 2000
take place in Finland (Im et al., 2019). These numbers do not include the possible impacts of above mentioned health
impacts of nanoparticles.
In recent years, instrument development has enabled the detection of aerosol particles with diameters between 1 and 3 nm
(Vanhanen et al., 2011), which we here refer to as sub-3 nm particles. This had made it possible to study the very
beginning of NPF, which starts with the formation of sub-3 nm particles. NPF events are favored in specific
meteorological conditions, for example, high solar radiation and low relative humidity, as well as an abundance of low-
volatile gaseous precursors (Hussein et al., 2008; Kerminen et al., 2018; Salma et al., 2011; Wonaschütz et al., 2015).
One of the known gaseous precursors, which plays an important role in NPF, is sulfuric acid (SA). SA is an oxidation
product of sulfur dioxide ($SO_2$), which is primarily emitted from anthropogenic processes related to fuel combustion, for
example from traffic. Previous studies have shown that SA is also directly (Arnold et al., 2012; Rönkkö et al., 2013) and
indirectly via solar radiation (Olin et al., 2020) emitted by traffic. In many locations, SA concentration is one of the critical
factors determining whether NPF occurs (Kuang et al., 2008; Ripamonti et al., 2013; Wang et al., 2011).
In addition to potential NPF events, traffic is a significant source of the sub-3 nm particles (Hietikko et al., 2018; Rönkkö
et al., 2017). The sub-3 nm particles are directly emitted from vehicle exhaust and brake wear (Nosko et al., 2017) or
formed in the exhaust plume from the nucleating gaseous components  (Rönkkö and Timonen, 2019). On the other hand,
the concentrations from the road emissions decrease fast while moving away from a road due to dispersion (Pirjola et al.,
2006). Especially, the concentration of the sub-3 nm particles is additionally reduced by condensation and coagulation
(Kangasniemi et al., 2019). Rönkkö et al. (2017) showed that the sub-3 nm particles represent 20-54% of the particle
population in a 'semiurban' roadside environment in Helsinki. Kontkanen et al. (2017) analyzed the sub-3 nm particle
concentrations in different environments and concluded that the sub-3 nm particle concentrations are higher in locations
influenced by anthropogenic emissions. Detailed analysis has linked the sub-3 nm particles to traffic activity and traffic
emissions at the street canyon in Helsinki (Hietikko et al., 2018). Traffic emissions do not only contain particles but also
SA, volatile organic compounds, and trace gases for example carbon dioxide ($CO_2$) and nitrogen oxides ($NO_x$), which are
commonly used as traffic markers. Generally, the role of traffic in urban air quality and its effects on human health are
still not well understood.
Previously, particle size distributions have been measured in different urban environments, such as London (Bousiotis et
al., 2019; Harrison et al., 2019; Hofman et al., 2016), Stockholm (Mårtensson et al., 2006), Innsbruck (Deventer et al.,
2018), Los Angeles (Zhu et al., 2002), Beijing (Zhou et al., 2020), Shanghai (Xiao et al., 2015), and Helsinki (Hussein et
al., 2006; Ripamonti et al., 2013). In the Helsinki area, the focus of the research has been either on NPF (Hussein et al.,
2008, 2009) or the primary particle emissions (Hietikko et al., 2018; Ripamonti et al., 2013; Rönkkö et al., 2017). These
approaches leave an open question about the relative contribution of each source to the particle population. To answer
this question, we conducted simultaneous measurements at two close-by stations in the Helsinki area: at the street canyon
and at the urban background station. For the first time, particle size distribution in a diameter range from 1 to 800 nm as
well as the concentrations of precursor gases were simultaneously measured at two nearby stations in Helsinki. In this





article, we present the results of these measurements and compare the particle size distributions and their variation at
these two stations. Specifically, we develop and apply a new method to determine the relative contributions of NPF and
traffic to the sub-3 nm particle population in different urban environments.

## 2. Methods

### 2.1. Measurement stations

We performed measurements at two different stations in Helsinki, Finland, during April-June 2018. The first one is
Helsinki Region Environmental Services (HSY) air quality station, which is placed in a street canyon (Mäkelänkatu street,
approximately 28 000 vehicles per workday) and represents a busy street environment (Kuuluvainen et al., 2018). The
second one, the Station for Measuring Ecosystem-Atmosphere Relations (SMEAR III), is located within 900 m north-east
of HSY site, and it is classified as an urban background station (Fig. 1) (Järvi et al., 2009). The sites are separated by
buildings, a botanic garden, and a small deciduous forest. The SMEAR III is located on a hill, approximately 12 m above
the nearest busy road (Hämeentie street). The SMEAR III is separated from Hämeentie by a 150 m band of a deciduous
forest. Apart from the forest, in the SMEAR III surrounding are also buildings, parking lots, and small vegetation (Järvi
et al., 2009). In this article, the two measurement stations are called 'street canyon' and 'background', respectively.

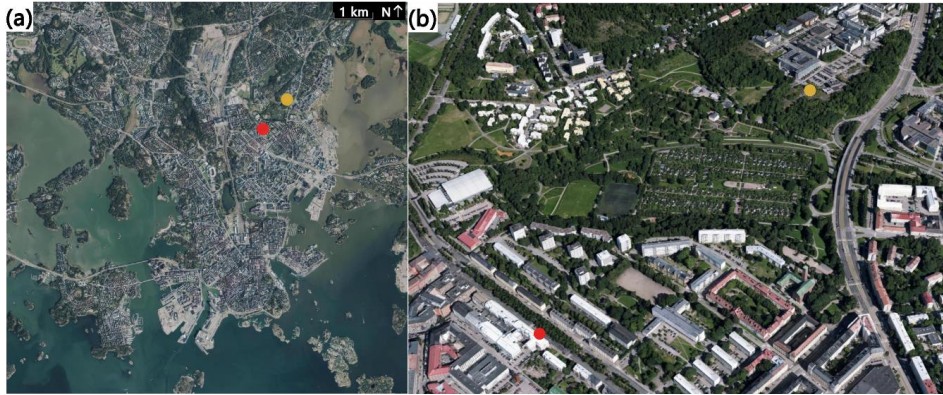

Figure 1. Aerial photography (a) and 3d model (b) of stations: street canyon (red) and background (yellow). The
photograph and the model were provided by The City of Helsinki map service (CC BY 4.0).

### 2.2. Measurement period and comparison of data from sites

We measured particle number size distribution, trace gas and SA concentrations at both stations during the period 27 April
2018 – 5 June 2018. An overview of instruments used during this campaign is presented in Table 1 indicating the total
running time at each station. The detailed working time for each instrument is shown in Table A1. Most of the analysis
was conducted separately for workdays and days free of work, i.e. weekends and holidays (1 May 2018 and 10 May
2018), which are for simplicity just referred to as 'weekends' in this article. When comparing particle concentrations from
two stations, we analyze times when all the instruments measuring particles were performing at each site. This resulted
in 120 and 101 hours of measured particle concentration during weekends and workdays, respectively, at the street canyon
site. At the background station, it resulted in 217 hours of observed particle concentration during weekends and 398 hours
during workdays. In addition, we present separately a few case studies, where the overlapping data obtained





simultaneously from two stations are analyzed in more detail. Oppositely to the analysis of particle concentrations,
sulfuric acid (SA) concentration was studied only for the overlapping period of measurements at two stations resulting in
167 and 329 hours measured during weekends and workdays, respectively. Condensation sink (CS) was studied from the
same time frame as SA.
**2.3. Particle size distribution measurement**
A wide range of particle size distribution was obtained by combining several instruments: a Particle Size Magnifier (PSM)
(Vanhanen et al., 2011), an Ultrafine Condensation Particle Counter (UCPC), and a Condensation Particle Counter (CPC)
(Kangasluoma and Attoui, 2019) as well as a Differential Mobility Particle Sizer (DMPS) (Wiedensohler et al., 2012).
The measured size ranges and working hours of these instruments are presented in Table 1.
The PSM technique contains a pre-conditioner, that activates the smallest particles and grows them up to 90 nm, and a
CPC (Vanhanen et al., 2011). The minimum size of activated particles depends on the diethylene glycol supersaturation
in the pre-conditioner (Lehtipalo et al., 2014). Altering the supersaturation condition allows varying the minimum size of
activated and measured particles between 1 and 3 nm. The PSM can be used to measure particle concentrations in three
different modes: fixed, step, and scanning. In the fixed mode, supersaturation and the minimum size of the measured
particles are constant. Data obtained by measuring in the fixed mode have a high temporal resolution (1 s), and therefore
it is mainly used in a very rapidly changing environment such as a busy street. In the step mode, supersaturation and the
lowest size of measured particles oscillate between three set values. This allows analyzing particle size distribution in the
range from 1 to 3 nm. On the other hand, data obtained from step mode measurements have a lower temporal resolution
(3 min). Adjusting the time of every supersaturation measurement allows minimizing the uncertainties related to the rapid
changes in the analyzed environment. In the scanning mode, supersaturation gradually changes between two set values.
The scanning mode enables choosing the size bins when inverting the raw data to a size distribution. However, the
temporal resolution of scanning mode is the lowest of all the modes (4 min). In the scanning mode, we assume that particle
concentration stays constant during each scan, thus this mode cannot be used in a rapidly changing environment. In this
study, PSM was operated in fixed and step modes. At both stations, PSMs working in the fixed mode measured
concentrations of particles with sizes larger than 1.2 nm.
Condensation particle counters enlarge particles by condensation of supersaturated condensable vapors. Once particles
reach a size sufficient for optical detection, they are counted by the optical particle counter. In this study, we use two
models of butanol-based CPCs, from which one measures particle concentration of particles with sizes larger than 3 nm,
while the other one counts particles with sizes larger than 7 nm. For simplicity, we call them UCPC and CPC, respectively.
Differential mobility particle sizer (DMPS) consists of a differential mobility analyzer (DMA) and a CPC. The DMA
classifies charged particles according to their mobility in an electric field. By incrementally stepping the voltage applied
to the central rod of the DMA, particles of lower mobility can be classified by the DMA and further quantified by the
CPC. During an 8 minute cycle, DMPS monitors the size distribution of particles with a diameter between 6 and 800 nm.
The size distribution obtained from DMPS measurements was used to study the loss rate of vapors due to condensation
on existing particles, i.e. CS (Kulmala et al., 2012). DMPS data from the background station was also utilized to identify
NPF events based on the method proposed by Dal Maso et al. (2005).





All instruments were corrected for diffusion losses in their inlets except the UCPC measuring at the background station,
in which the core sampling technique (Fu et al., 2019) was used.
Sub-3 nm particle concentration was determined by subtracting concentration measured by the UCPC from concentrations
measured by the PSM working at the fixed mode. The nucleation mode (3-25 nm) was computed by adding concentration
measured by DMPS with diameters of 7-25 nm to the difference of concentration measured by CPC and UCPC. Particle
concentrations measured by DMPS with diameters of 25-100 nm and 100-800 nm were considered the Aitken and
accumulation mode concentrations. The size range for the Aitken mode corresponds to the range of mean particle size in
the typical soot mode of vehicle exhaust (Rönkkö and Timonen, 2019). At the street canyon site, gaps in the UCPC data
were filled with the concentration of particles larger than 3 nm obtained from the PSM measuring in the step mode. At
the background station, UCPC data from 29 May to 4 June were corrected by checking that the ratios of concentrations
measured with the PSM and UCPC as well as UCPC and CPC agree in the night-time. All the measured size ranges
correspond to the mobility diameter of particles.
Table 1. Overview of the main instruments used at the street canyon and background station during the campaign.

| Instrument | Description | Working time at street canyon [h] | Working time at background station [h] |
|---|---|---|---|
| PSM (fixed mode) | Concentration of particles larger than 1.2 nm | 246 | 766 |
| PSM (step mode) | Particle size distribution in the range of 1-3 nm | 833 | - |
| UCPC | Concentration of particles larger than 3 nm | 548 | 670 |
| CPC | Concentration of particles larger than 7 nm | 750 | 728 |
| DMPS | Particle size distribution in the range of 6-800 nm | 840 | 821 |
| CI-APi-TOF | Concentration and chemical identification of vapor molecules and molecular clusters in size range approximately 0.1-1 nm. In this study only sulfuric acid concentration is utilized. | 501 | 776 |
| $SO_2$ analyzer | Concentration of $SO_2$ on ppb level | 452 | 937 |
| $CO_2$ analyzer | Concentration of $CO_2$ on ppm level | 771 | 248 |
| NO/$NO_x$ analyzer | Concentration of NO/$NO_x$ on ppm level | 937 | 937 |

### 2.3.1. Uncertainties of sub-3 nm particles measurement
Measuring the concentration of particles smaller than 3 nm contains noteworthy uncertainties mainly due to the effect of
chemical composition and charging state of particles on the cutoff size of PSM.
Particle detection efficiency in butanol counters (Wlasits et al., 2020) and the PSM techniques depend strongly on the
chemical composition of measured clusters. Experiments show that the difference between the cutoff diameter in PSM
for particles with different chemical composition can reach up to approximately 1 nm (Jiang et al., 2011; Kangasluoma
et al., 2014, 2016). This causes uncertainty of ± 0.5 nm for particles with unknown chemical composition. PSM calibrated
with particles with the same chemical composition as measured one would have a negligible offset (Kangasluoma et al.,





2015). In the urban environment, the chemical composition of particles is complex and evolving with time, thus this
uncertainty cannot be minimized in this research.
Uncertainty due to the charging state of particles is mainly affected by the discrepancy between the charging state of
measured particles and particles used for the calibration. PSMs were calibrated with negatively charged tungsten oxide
clusters by the method presented in Kangasluoma et al. (2015). However, the majority of particles measured in the urban
environment are likely electrically neutral (Yao et al., 2018). Since neutral particles are activated at a higher
supersaturation than charged particles, we probably underestimate the size of measured particles (Kangasluoma et al.,
2016, 2017; Winkler et al., 2008). Experiments indicate a 0.1-0.5 nm increase in the activated diameter of neutral particles
in PSM in comparison to the charged ones (Kangasluoma et al., 2016, 2017). When the effect of charge on the measured
particle population is unknown, increasing the cutoff diameter in PSM by 0.3 nm will reduce the uncertainty of the state
of charge to ± 0.2 nm (Kangasluoma and Kontkanen, 2017).
Additionally, meteorological conditions such as humidity can affect the cutoff size of PSM and CPC (Kangasluoma et
al., 2013; Tauber et al., 2019).
Lastly, the non-ideal efficiency curve, used for determining the cutoff diameter, makes it possible to sample particles
smaller than the cutoff size. When the relative contribution of sub-3 nm particles to the total particle population is high,
the uncertainties of cutoff diameter or the shape of the efficiency curve can affect the total concentration measured by
PSM and CPC (Kangasluoma and Kontkanen, 2017).
**2.4. Sulfuric acid measurement**
Sulfuric acid (SA) concentration was monitored with a high-resolution chemical ionization atmospheric pressure interface
time-of-flight mass spectrometer (CI-APi-TOF) with nitrate ($NO_3^-$) as a reagent ion. CI-APi-TOF technique contains a
chemical ionization source (CI) and an atmospheric pressure interface (APi) coupled with a time-of-flight mass
spectrometer (TOF). Chemical ionization is a soft ionization technique in which the reagent ion reacts with analyzed
compounds and charge them. $NO_3^-$ is a reagent ion used for the detection of gaseous SA in ambient air (Eisele and Tanner,
1993; Jokinen et al., 2012; Mauldin et al., 1999). $NO_3^-$ or its cluster with nitric acid ($HNO_3NO_3^-$) reacts via proton transfer
reaction with sulfuric acid creating a cluster detectable by the APi-TOF technique (Jokinen et al., 2012). A detailed
description of nitrate-based CI method used as a pre-treatment for APi-TOF technique is presented by Jokinen et al.
(2012). The atmospheric pressure interface guides ionized compounds through three stages of lowering sample pressure
to the time-of-flight region. A TOF mass spectrometer separates and detects analyzed compounds by their mass-to-charge
ratios. A detailed description of the APi-TOF technique is described by Junninen et al. (2010).
Due to the uncertainty of the rate of the reaction between gaseous sulfuric acid and the nitrate ion, the CI-APi-TOF needs
to be calibrated by taking the wall losses of SA inside the instrument and the flow conditions of the ion source into
consideration (Kürten et al., 2012; Viggiano et al., 1997). Calibration of CI-APi-TOF was done before the campaign,
based on the method proposed by Kürten et al. (2012). The SA concentration was calculated from Eq (1) (Jokinen et al.,
194    2012).


$$[SA] = C \cdot \frac{CR_{97} + CR_{160}}{CR_{62} + CR_{125} + CR_{188}} \qquad (1)$$



where [SA] is SA concentration, C is the calibration coefficient and $CR_M$ is a count rate of an ion with a mass M in
amu.
The SA zero level concentration, determined by measuring filtered air, was subtracted from the measured concentrations.
Uncertainties of absolute concentration measured by CI-APi-TOF are in the order of 50%, while the uncertainties of
relative changes in the concentration are smaller than 10% (Ehn et al., 2014).

### 2.5. Other instrumentation

Nitric oxide (NO), $CO_2$, $SO_2$, and $NO_x$ were additionally measured during this campaign (Table 1). To complement these
measurements, we used continuous measurements performed at both stations. These include meteorological parameters,
ozone ($O_3$) concentration, ion size distribution, and black carbon (BC) concentration. All the instruments used are listed
in Table A2.

### 2.6. Estimation of the relative contribution of NPF and traffic to sub-3 nm particles

To estimate the influence of traffic and NPF on the sub-3 nm particle population, we analyzed the correlation between
sub-3 nm particles and $NO_x$ concentrations as well as between sub-3 nm particles and SA concentrations. $NO_x$
concentration was used as a traffic marker (Olin et al., 2020) while SA concentration was used as an NPF marker (Sipila
et al., 2010). Bivariate fittings (Cantrell, 2008; Williamson, 1968; York, 1966) were conducted on the common logarithms
of sub-3 nm particles and SA when $NO_x$ concentration was low to estimate sub-3 nm particles concentration originating
from NPF. Correlation between common logarithms of sub-3 nm particles and $NO_x$, when SA concentration was low,
was used to estimate sub-3 nm particle concentration originating from traffic. Equations used for calculating sub-3 nm
particles emitted by traffic and NPF as well as their relative contributions to the particle population are presented in the
Appendix (Eq. A1-A6).

## 3. Results and Discussion

### 3.1. NPF event classification

The results of the NPF event classification at the background station for the studied period is shown in Table 2. The
examples of an event, non-event, and undefined class are shown in Fig. S1. The overall frequency of NPF event days was
12.5%; 21% of weekends and 8% of workdays were classified as events. Due to nucleation mode particles originating
from local sources, the majority of days were classified as undefined.
Table 2. NPF event classification at the background station for the period 27 April 2018 – 5 June 2018. Results are
presented separately for a full campaign, weekends, and workdays.

| Class | Date | $Freq_{campaign}$ | $Freq_{weekends}$ | $Freq_{workdays}$ |
|---|---|---|---|---|
| Event | 5.05, 7.05, 10.05, 13.05, 28.05 | 12.5% | 21% | 8% |
| Non-event | 29.04, 1.05, 6.05, 11.05-12.05, 15.05, 21.05, 26.05, 3.06 | 22.5% | 43% | 11% |
| Undefined | 27.04-28.04, 30.04, 2.05-4.05, 8.05-9.05, 14.05, 16.05-20.05, 22.05-25.05, 27.05, 29.05-2.06, 4.06-5.06 | 65% | 36% | 81% |

### 3.2. Particle size distributions



Figure 2 presents the median particle size distributions at both stations during workdays and weekends at different times
of the day. The shape of the size distributions for Aitken (25-100 nm) and accumulation mode (100-800 nm) particles is
quite similar at the two sites, but the concentrations are higher at the street canyon, as discussed later in this section. At
the street canyon site, the concentration of particles in a nucleation mode (3-25 nm) has a decreasing trend with an increase
of particle size. During the morning, noon, and afternoon, the nucleation mode has a peak below 10 nm, which is
characteristic of primary emitted particles from traffic (Rönkkö and Timonen, 2019). Similarly, this peak is observed for
nucleation mode particles at the background station during the morning. At the background station, during the night and
afternoon, the concentration of particles in nucleation mode increases with increasing particle diameter. During noon the
nucleation mode has a peak above 10 nm, likely linked to an NPF event. A sudden change in concentrations of particles
with a diameter below and above 7 nm at the background station can be associated with the uncertainty of measurement
with different instruments particles smaller than 10 nm (Kangasluoma et al., 2020). The shape of the particle size
distribution at the background station is somewhat different from the average size distribution measured at the same
location in the years 1997-2003 (Hussein et al., 2004). Hussein et al (2004) found that during spring the size distribution
of 8 and 400 nm particles reaches the maximum concentration in the nucleation mode, while in our study concentration
of particles within the same size range has a maximum in the Aitken mode. This difference could be explained by a higher
contribution of NPF to the average size distribution determined by Hussein et al (2004). At the street canyon site, the
shape of the size distribution of larger than 5 nm particles observed in our study is quite similar to the one measured at
the same location in May 2017 (Hietikko et al., 2018). In 2017, the particle concentration was observed to reach a
maximum for particles around 5 nm (Hietikko et al., 2018), while in our case the highest concentration during daytime is
reached for particles smaller than 3 nm.
Focusing on the smallest particles (Fig. 3), we observe that at the street canyon the median concentration of sub-3 nm
particles is up to $2.4*10^4$ cm$^{-3}$ higher than at the background station (Fig. 3c). The concentration of sub-3 nm particles is
higher at the street canyon site regardless the particle loss due to coagulation scavenging being twice as high as at the
background station (discussed later on in this section). At the street canyon, two traffic-related peaks are observed during
the morning (6:00-8:00) and afternoon hours (15:00-17:00) on workdays. These peaks correspond to the increase of NOx
concentration at the street canyon site during workdays (Fig. S3a). During weekends, there is no morning peak and the
afternoon peak occurs earlier (14:00-17:00). The level and the diurnal variations of sub-3 nm particles at the street canyon
is similar to observations at the same site in May 2017 by Hietikko et al. (2018). They found that sub-3 nm particles
followed the pattern observed in numbers of vehicles at Mäkelänkatu street. However, in 2017 the morning peak in
sub-3 nm particles was shorter and the afternoon peak started one hour earlier. At the background station, the diurnal
variation of sub-3 nm particle concentration has a maximum around noon both during weekends and workdays.
Nevertheless, a sharp increase of sub-3 nm particles concentration is observed in the morning (6:00) during workdays.
Morning raise of sub-3 nm particle concentration at the background station during workdays corresponds to a peak of
NOx concentration (Fig. S3b), which suggests the contribution of traffic emissions. The absence of clearly visible
traffic-related peaks at the background station could be caused by the 150 m distance of the site uphill from the nearest
road as well as the separation of the road and the station by the forest. The midday maximum of sub-3 nm particles
concentration is likely related to NPF. However, Kontkanen et al. (2017) showed that at the background station the starting
time of a sub-3 nm particle concentration increase varies hardly throughout the year. This means that the increase in
sub-3 nm particles concentration is independent of the sunrise, respectively solar radiation. This suggests the partial
influence of traffic on the observed peak.



The difference between the stations (Fig. 3c) shows that median sub-3 nm particle concentrations during the rush hours
in the street canyon site are clearly higher than at the background station throughout, roughly by a factor of 5. However,
the concentrations are slightly higher also during other times of the day, which shows the influence of the continuous
traffic emissions at the street canyon site.

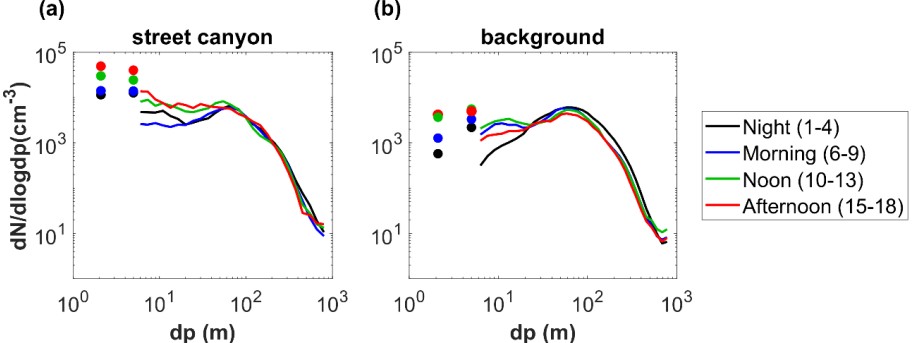

Figure 2. Median size distribution (a) at the street canyon and (b) at the background station. The colors indicate different
periods of the day: night (1:00-4:00 LT, black), morning (6:00-9:00 LT, blue), noon (10:00-13:00 LT, green), and
afternoon (15:00-18:00, red). Median size distribution was determined by DMPS (particles with sizes between 6-800 nm)
marked with solid lines in the figure, UCPC and CPC (3-7 nm), and PSM and UCPC (1-3 nm) marked with dots. This
figure with the linear y-axis can be found in Supplementary material (Fig. S2).

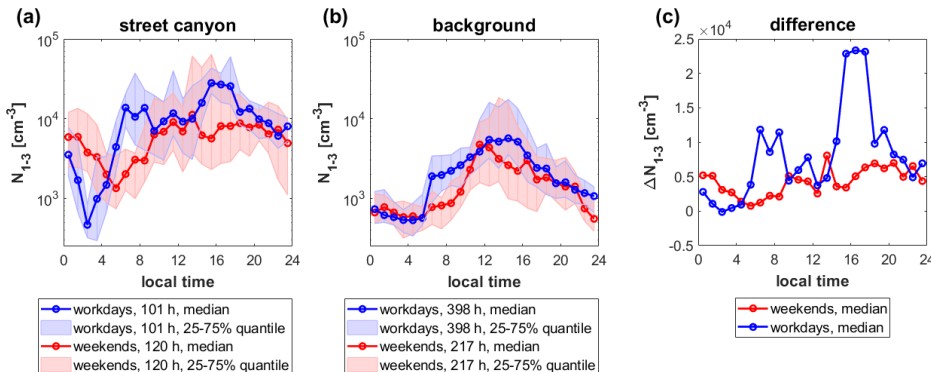


Figure 3. The diurnal variation of sub-3 nm particle concentration during weekends (red) and workdays (blue) (a) at the
street canyon and (b) at the background station, and (c) the difference between median sub-3 nm particles concentration
at the street canyon (a) and at the background station (b). The median diurnal variation is shown as a solid line with
markers; the 25th and 75th percentile ranges are presented as shaded areas.
The diurnal variation of nucleation mode particle concentration is similar to that of sub-3 nm particles at both stations
(Fig. 4). During workdays, we can see traffic-related peaks, which are less pronounced on weekends. The concentrations
of nucleation mode particles in the street canyon are $10^3$-$3.9\times10^4$ cm$^{-3}$ higher than at the background station during the
daytime. Traffic peaks are also pronounced in the diurnal cycle of Aitken mode particles measured on workdays in the
street canyon. Concentrations of Aitken mode particles at the street canyon site are up to $5\times10^3$ cm$^{-3}$ higher than at the





background station on workdays. During nighttime and weekends, concentrations of Aitken mode particles are similar at
both stations, which suggests a similar origin of these particles. The diurnal variation of Aitken mode particle
concentration at the background station is similar during workdays and weekends, however, during daytime
concentrations are higher on workdays. At street canyon during workdays, we can observe traffic-related peak in Aitken
mode particles, which are absent during weekends. Accumulation mode particle concentrations during workdays and
weekends are comparable at each of the stations (Fig. S4). During daytime accumulation mode particles reached higher
concentrations at the street canyon than at the background station, which causes a difference of roughly a factor of two
between condensation and coagulation sinks at the sites (Fig. 5). Accumulation mode particle concentration is almost
constant during the whole day.
Overall, the influence of traffic on the particle population at the street canyon is clearly visible for sub-3 nm, nucleation
mode, and Aitken mode particles, while the accumulation mode is only slightly influenced by traffic. The particle
concentrations at the background station are also influenced by traffic, but not as strongly as at the street canyon station.
At the background station, the influence of traffic can be observed only for sub-3 nm and nucleation mode particles. These
results suggest that sub-3 nm and nucleation mode particle concentrations in the urban environment are mainly influenced
by local sources, while the accumulation mode particle concentrations are mostly dominated by transport from non-local
sources. Whether the Aitken mode is primarily dominated by local or non-local sources depends on the analyzed location.

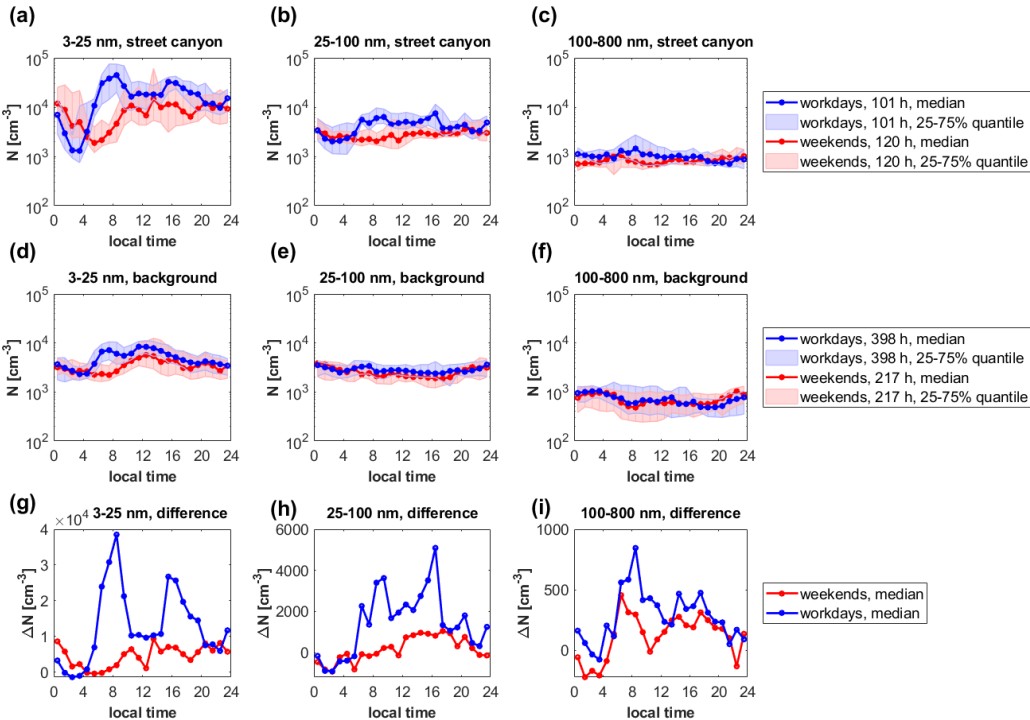


Figure 4. Diurnal variations of nucleation (3-25 nm), Aitken (25-100 nm), and accumulation (100-800 nm) modes particle
concentration during weekends (red) and workdays (blue) measured at the street canyon (top) and background station
(middle) as well as the difference between the street canyon site and the background station concentrations (bottom). The



median diurnal variations are shown as solid lines with markers; the 25th and 75th percentile ranges are presented as
shaded areas. This figure with the linear y-axes can be found in Supplementary material (Fig. S4).

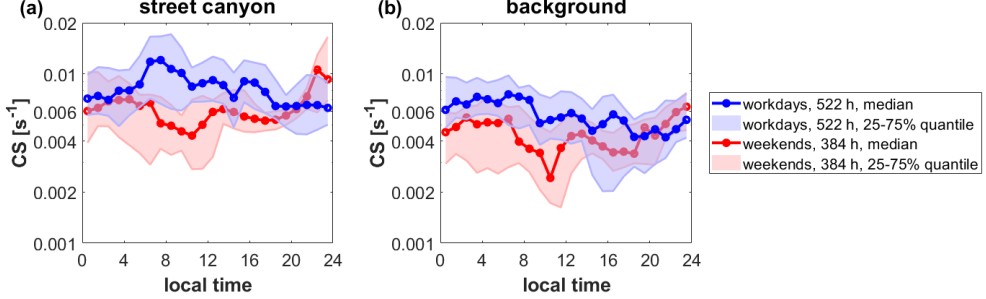


Figure 5. The diurnal variation of CS during weekends (red) and workdays (blue) (a) at the street canyon and (b) the
background station. The median diurnal variation is shown as a solid line with markers; the 25th and 75th percentile range
are presented as a shaded area.
**3.3. Sulfuric acid**
SA concentration had a clear daytime maximum at both sites (Fig. 6). The only difference in SA concentration between
workdays and weekends at each site was observed during midday (13:00-15:00) when SA concentration reached higher
values during weekends than weekdays. During weekends, the median SA concentration at the background station had a
maximum of $1.1*10^7$ cm$^{-3}$ while during workdays, it reached only $4.6*10^6$ cm$^{-3}$. A similar pattern is observed at the street
canyon station with maximum concentrations of $6.9*10^6$ cm$^{-3}$ and $3.6*10^6$ cm$^{-3}$ during weekends and weekdays,
respectively. This difference is likely linked to a bigger fraction of NPF events days during analyzed weekends than
workdays (Table 2). Daytime median SA concentrations are slightly higher at the background station than at the street
canyon (Fig. 6, Fig. S5), which is probably caused by higher CS at the street canyon site (Fig. 5). In contrast to daytime,
nighttime median SA concentrations are an order of magnitude higher at the street canyon site than at the background
station. This difference should not be caused by different instrumental background, as we corrected SA data with zero
measurements. High nighttime SA concentration at the street canyon could be caused by direct emission of sulfuric acid
from traffic (Arnold et al., 2012) or nighttime SA formation (Guo et al., 2020). SA concentrations at the street canyon
site are slightly lower than concentrations measured one year earlier by Olin et al. (2020).

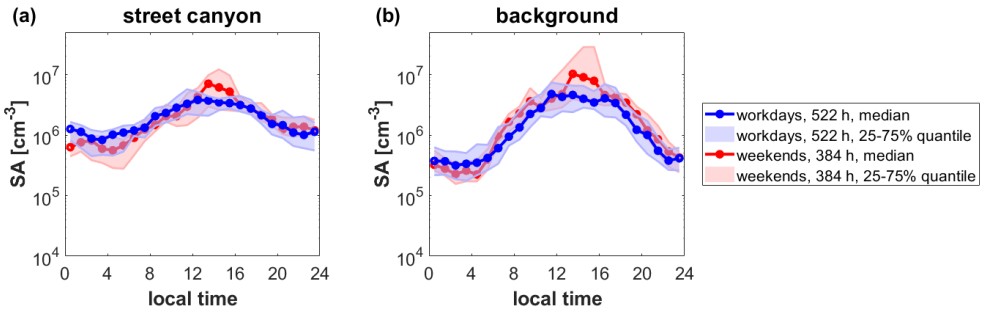


Figure 6. The diurnal variation of sulfuric acid (SA) concentration (a) at the street canyon and (b) at the background



station. The median diurnal variation is shown as a solid line with markers; the 25th and 75th percentiles are presented
as a shaded area.
**3.4. Case studies**
To understand the behavior of sub-3 nm particles on a shorter time scale, we analyzed two periods, when the particle
concentration data are available from both sites: Saturday 5 May 2018-Sunday 6 May 2018 and Tuesday 8 May 2018
09:00-Wednesday 9 May 2018 15:00. The first investigated case is a weekend starting with an NPF event (Fig. 7a,b). The
second case contains typical workdays (Fig. 7c,d), which are classified as undefined days in NPF event classification.
Supporting information about atmospheric conditions, trace gas concentrations, black carbon (BC) concentration, and CS
during the analyzed cases are presented in Fig. S6-S8. In both studied cases, sub-3 nm particles and SA concentrations
follow each other closely at the background station (Fig. 8). Oppositely, at the street canyon site, there are many traffic-
related sub-3 nm particles peaks, which often do not have their counterparts in SA time-series. This suggests that the
majority of SA is not originating from direct emissions from traffic. Our analysis supports studies showing that sub-3 nm
particles are not only formed by clustering of atmospheric vapors, but it is also directly emitted from traffic (Hietikko et
al., 2018; Rönkkö et al., 2017; Rönkkö and Timonen, 2019). The pattern of SA time-series is similar at both stations, but
SA concentrations are lower at the street canyon. The highest sub-3 nm particles and SA concentrations during both cases
were measured at each site during the NPF event. The relation between sub-3 nm particles measured at the street canyon
and background station during these case studies is presented in Fig. S9. During the NPF event (Fig. 8a,b), sub-3 nm
particles concentration at the background station is almost a factor of two higher than at the street canyon site. However,
nearly simultaneous to the highest peak in sub-3 nm particles and SA concentrations, a peak in particle concentrations
across the modes (Fig 7b) as well as in $SO_2$ concentration (Fig. S6) is observed at the background station. This seems not
to be a feature of a regional NPF event but could be a plume from e.g. a ship or a coal-fired power plant in Helsinki,
which happens to be more efficiently transported to the background station than to the street canyon. This illustrates the
interplay of various types of sources on the aerosol concentrations, regional NPF events, local traffic sources, and nearby
point sources. It should be kept in mind that this case occurs during the weekend when the traffic volumes are lower and
daily patterns of traffic rate differ from weekdays, and thus on average the influence of traffic is expected to be more
pronounced.

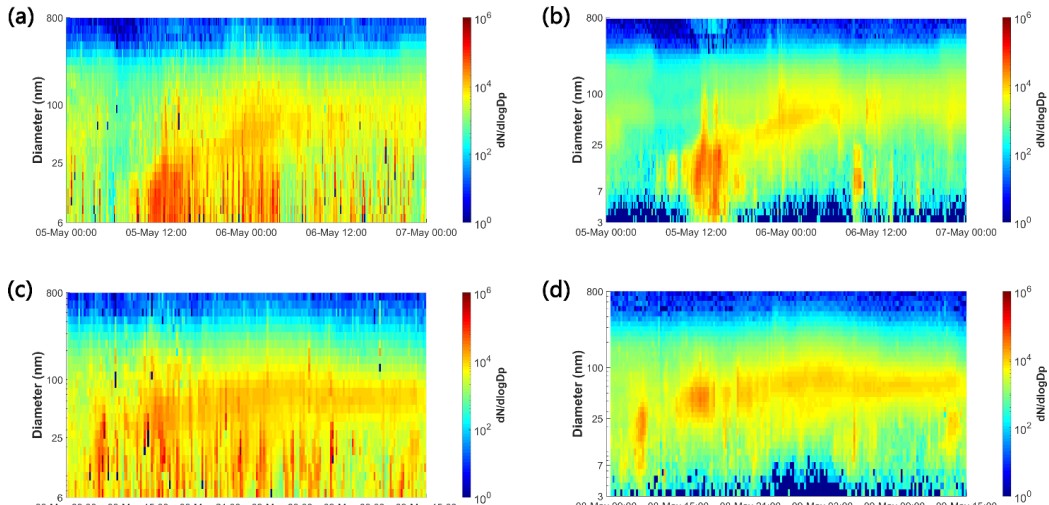

Figure 7. Time series of particle size distribution at the background station (b, d) and at street canyon site (a, c)
measured by DMPS for periods of 5 May 2018-7 May 2018 LT (a, b) and 8 May 2018 09:00-9 May 2018 15:00 LT
(c, d).

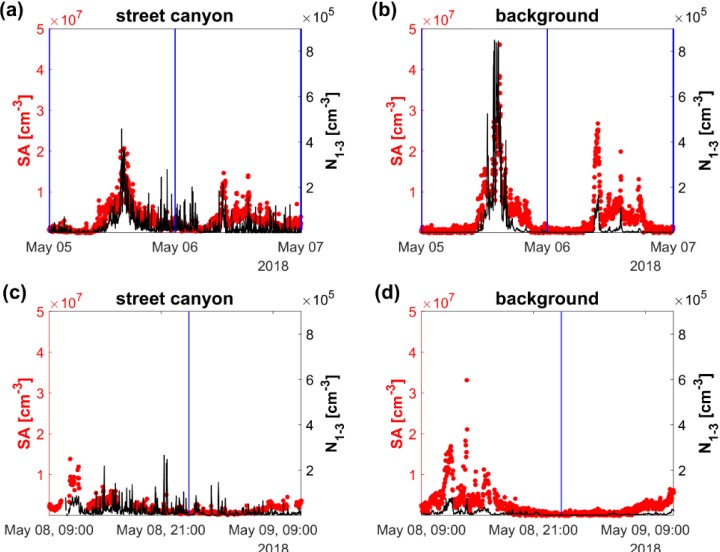

Figure 8. Time series of sub-3 nm particles (black) and SA (red) concentrations at the street canyon  (a, c) and at the
background station (b, d) during 5 May 2018-7 May 2018 LT (a, b) and 8 May 2018 09:00-9 May 2018 15:00 LT (c, d).
Vertical blue lines indicate midnights. This figure with the logarithmic y-axis can be found in Supplementary material
(Fig. S10).







362 **3.5. Regression analysis**

363 To investigate in detail the contribution of different sources to the sub-3 nm particles population at both sites, we analyzed

364 correlations of different variables with sub-3 nm particle concentration (Table 3). Sub-3 nm particle concentration at the

365 background station correlates best with the concentrations of $SO_2$ (R = 0.64 on workdays and R = 0.46 on weekends) and

366 its oxidation product SA (R = 0.66 on workdays and R = 0.66 on weekends), which is a common precursor of NPF.

367 Sub-3 nm particle concentration at the street canyon correlates best with NO (R = 0.65 on workdays and R = 0.57 on

368 weekends) and $NO_x$ concentrations (R = 0.62 on workdays and R = 0.54 on weekends). Generally, at the street canyon

369 site, high correlations are observed between sub-3 nm particles and species, that can be associated with emissions from

370 traffic: BC, $NO_x$, NO, $CO_2$. The correlation between sub-3 nm particles and SA is also positive and high but weaker than

371 at the background station, which is in agreement with results from the case studies. Overall, the correlation analysis

372 suggests that the sub-3 nm particles population at the street canyon site is more influenced by traffic than at the background

373 station. This is discussed more in the next section.

374 Table 3. Correlation between logarithmic values of sub-3 nm particle concentration and logarithmic values of other

375 variables during weekends and workdays at the street canyon and the background station. N shows a number of measured

376 points taken into analysis for each period. In the case of missing data for an analyzed parameter, the number of studied

377 points is shown next to the correlation parameter. All correlations presented in the table are statistically significant at a

378 significance level of α = 0.05. Correlations with a Pearson's correlation coefficient higher than 0.5 are marked in bold.

| Parameter | Street canyon site | | Background station | |
|---|---|---|---|---|
| | Workdays (N=552) | Weekends (N=696) | Workdays (N=2366) | Weekends (N=1467) |
| SA [#/cm$^3$] | 0.49 | **0.61** | **0.66** | **0.66** |
| BC [ng/m$^3$] | **0.60** | 0.33 (N=418) | 0.15 | 0.27 |
| CS [s$^{-1}$] | 0.29 | 0.24 (N=418) | -0.10 | 0.13 |
| NO [ppb] | **0.65** | **0.57** | 0.37 (N=1813) | 0.45 (N=892) |
| NO$_x$ [ppb] | **0.62** | **0.54** | 0.28 (N=2218) | 0.35 (N=1439) |
| O$_3$ [ppb] | -0.09 | -0.24 | 0.13 | 0.13 |
| CO$_2$ [ppm] | 0.38 | **0.56** | -0.24 (N=689) | -0.28 (N=603) |
| SO$_2$ ** [ppb] | - | - | **0.64** (N=2051) | 0.46 (N=1266) |
| RH* [%] | -0.43 | -0.47 | -0.15 | -0.11 |
| T* [°C] | 0.38 | 0.45 | 0.17 | 0.10 |
| WD* [°] | -0.24 | -0.04 | -0.20 | -0.25 |

379 *  Correlation calculated for logarithmic values of sub-3 nm particle concentration and standard values of the

380  parameter

381 **3.6. Estimation of NPF and traffic contribution to sub-3 nm particles**

382 Our results suggest that the sub-3 nm particle population at the urban background station is mainly influenced by particles

383 formed by atmospheric NPF, while at the street canyon site it is affected more by particles emitted by traffic. The

384 compounds that correlate best with sub-3 nm particles at each site, SA and $NO_x$, can be used as markers of NPF and traffic

385 emissions respectively. To quantify the influence of each process on sub-3 nm particle concentrations, we studied the





dependency between sub-3 nm particles, SA, and NO$_x$ concentrations at both sites (Fig. 9, Table S1). We made bivariate
fittings to common logarithms of NO$_x$ and sub-3 nm particles when the SA concentration was low and reversely we
analyzed common logarithms of SA and sub-3 nm particles when the NO$_x$ concentration was low. The bins were chosen
for fitting so that they were as similar as possible at both stations and contained enough data points. The slopes of the
bivariate fit to sub-3 nm particles and SA data for low NO$_x$ concentration is close to 1 at both stations (Fig. 9 a,b). At the
same time, the slope of the fit to sub-3 nm particles and NO$_x$ data for low SA concentration is considerably smaller at the
background station (0.64) than at the street canyon site (1.40) (Fig. 9 c,d). We investigated possible reasons for this
difference such as constant background (local source) of sub-3 nm particles at the background station or losses of sub-
3 nm particles due to CS, or particle growth. Analysis of the correlation between NO$_x$, SA, and total particle concentration
(Fig. S11), as well as the correlations between sub-3 nm particles, NOx, SA, and CS (Fig. S12-S13), implied that neither
particle growth out of the sub-3 nm size range nor varying CS can explain the difference in the slopes between stations.
One possible explanation could be a constant production of sub-3 nm particles at the background site, as a result of
clustering of low-volatile organic vapors (Rose et al., 2018). Comparing ion concentrations and sub-3 nm particles at the
background station indicates that the constant source of ions in the atmosphere cannot explain these high sub-3 nm particle
concentrations at the background station (Fig. S14). We should have in mind that compared ranges of NO$_x$ concentrations
are different at each station. Additionally, particle evaporation may affect the comparison.
Based on these bivariate fits, we estimated sub-3 nm particle concentration originating from NPF and traffic at the two
sites (Table 4). The analysis was done for the time when NOx, SA, and sub-3 nm particle concentrations were measured
at each station (Table A1, Fig. S15). The variability of estimated sub-3 nm particle concentration is high, and occasionally
estimated concentrations exceed the measured values of sub-3 nm particle while at other times estimated values are clearly
lower than the measured values (Fig. 10). However, our estimation captures the temporal variation of the sub-3 nm particle
concentrations adequately. We can conclude that during the daytime (6:00-20:00), a similar fraction of sub-3 nm particles
originate from traffic (53%) and NPF (47%) at the street canyon site. At the background station, the daytime sub-3 nm
particle population is dominated by particles originated from NPF (74%). During the nighttime (20:00-6:00), the influence
of both sources on the sub-3 nm particles population is almost equal at the background station. At the street canyon site,
sub-3 nm particles originate mainly from traffic (65%) during the night. Our estimation of the influence of traffic on sub-
3 nm particle population at background station (32%) and street canyon site (54%) agrees with the previous annual
estimation of sub-3 nm particles originating by traffic in Helsinki (6-84%) conducted by Olin et al. (2020). Overall, our
results are consistent with the fact that the regional NPF process occurs over a large spatial area, while traffic emissions
are local.
When discussing the estimated relative contribution of traffic and NPF to the sub-3 nm population, we should have in
mind that the conducted analysis does not take into account the origin of SA. Traffic can directly or indirectly emit SA,
thus traffic may influence SA concentration used for estimating sub-3 nm particles formed during NPF. This could cause
an underestimation of the relative contribution of traffic to the sub-3 nm population. Olin et al. (2020) estimated that
during May 2017, at typical workday noontime at the same street canyon site, the contribution of traffic to sub-3 nm
particles was approximately 85%. The difference between our results and the one presented by Olin et al. (2020) could
be partly caused by the influence of traffic to the SA concentration. However, Olin et al. (2020) calculated the traffic
contribution to the sub-3 nm particles for a typical workday, while most data (57.8%) from the street canyon site used for
our estimation was collected at the time free from work.



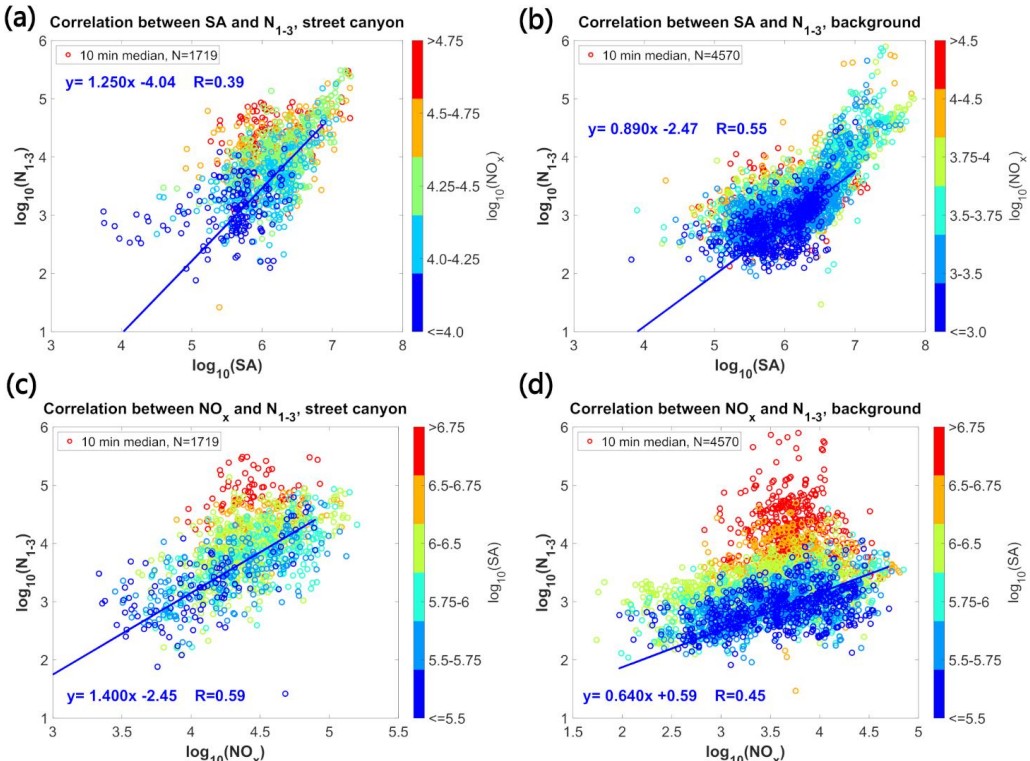

Figure 9. Correlation between the logarithm of SA and the logarithm of sub-3 nm particles colored by the logarithm of
$NO_x$ (a) at the street canyon and (b) at the background station, as well as the correlation between the logarithm of $NO_x$
and the logarithm of sub-3 nm particles colored by the logarithm of SA (c) at the street canyon and (d) at the background
station. Blue lines represent bivariate fit done to data with the logarithm of $NO_x$ smaller than 4 at street canyon site (a) or
3 at background station (b), or data with the logarithm of SA smaller than 5.5 at both stations (c,d). The parameters of the
fit are presented as an equation on the plot.



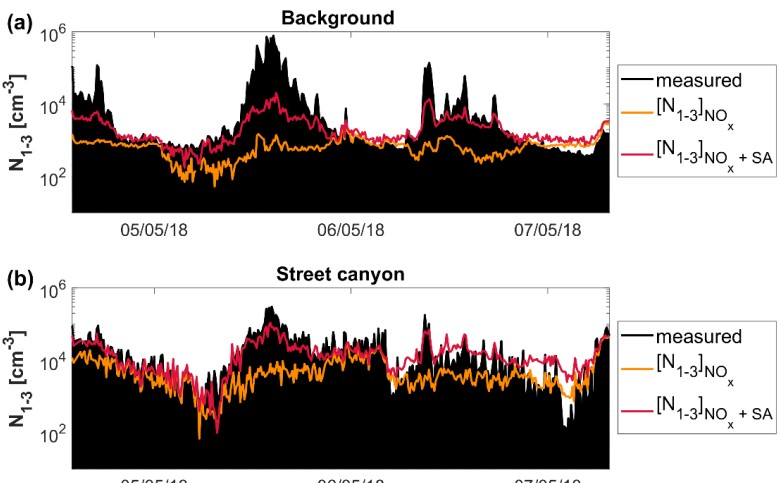

Figure 10. The time series of sub-3 nm particles concentration measured during 4 May 2018-8 May 2018 LT (black) and
estimated based on NOx concentration (orange) and $NO_x$ and SA concentrations (red) (a) at the background station and
(b) at the street canyon site.
Table 4. The percentiles of the relative contributions of traffic and NPF to estimated sub-3 nm particle concentrations at
the background station and the street canyon. $[N_{1-3}]_{NOx}$ and $[N_{1-3}]_{SA}$ present relative contribution of sub-3 nm particle
concentrations estimated based on $NO_x$ and SA concentrations, representing the contributions of traffic and NPF,
respectively. Equations used for calculating the relative contribution of each source are presented in Appendix
(Eq. A5-A6)

| | $[N_{1-3}]_{NOx}$ | | | $[N_{1-3}]_{SA}$ | | |
|---|---|---|---|---|---|---|
| **Percentile** | **25** | **50** | **75** | **25** | **50** | **75** |
| Daytime | | | | | | |
| Background [%] | 9 | 26 | 30 | 70 | 74 | 91 |
| Street canyon [%] | 26 | 47 | 70 | 30 | 53 | 74 |
| Nighttime | | | | | | |
| Background [%] | 41 | 67 | 75 | 25 | 33 | 59 |
| Street canyon [%] | 38 | 65 | 86 | 14 | 35 | 62 |
| All campaign | | | | | | |
| Background [%] | 14 | 43 | 57 | 43 | 57 | 86 |
| Street canyon [%] | 32 | 54 | 78 | 22 | 46 | 68 |

**4. Conclusions**
In this study, for the first time, the particle size distribution in a diameter range from 1 to 800 nm was analyzed at two
close-by stations in Helsinki. We found that the influence of traffic on particle number concentrations at the street canyon
is most visible for sub-3 nm, nucleation mode, and Aitken mode particles, while at the background station the influence



of traffic is clear only for sub-3 nm and nucleation mode particles. Sub-3 nm and nucleation mode particles are influenced
by local sources, especially traffic, while accumulation mode particles are dominated by long-range transportation from
non-local sources. Whether Aitken mode particles are more influenced by local or non-local sources, depends on the
location. At the background station, Aitken mode is largely dominated by non-local sources, while at the street canyon
site this mode is influenced by traffic.
We observed a very similar pattern in diurnal variation of SA concentration at both stations. Daytime SA concentrations
were slightly higher at the background station, likely due to a lower condensation sink than at the street canyon site.
During the nighttime, SA concentration was almost an order of magnitude higher at the street station. High nighttime
concentration at the street canyon site is probably caused by two simultaneous processes: direct SA emission from traffic
and nighttime SA formation.
Additionally, we performed two case studies, in which we analyzed the variation of SA and sub-3 nm particles on a short
time scale. Our study supports previous research showing that sub-3 nm particles include direct emissions from traffic.
During the NPF event on 5 May 2018, sub-3 nm particle concentrations at both sites were the highest, and traffic
contribution to the total sub-3 nm particle concentration at the analyzed stations was small.
Furthermore, we analyzed the relation of sub-3 nm particles with trace gases and meteorological variables. We observed
that sub-3 nm particles at the background station are mainly related to $SO_2$ and SA, while the sub-3 nm particle population
at the street canyon can be associated with components linked to traffic emissions (BC, $NO_x$, NO, and $CO_2$). Based on
these observations, we developed a new method to estimate the relative contributions of traffic and NPF to sub-3 nm
particle concentration at nearby urban sites. The relative impacts of traffic and NPF on the sub-3 nm particles in the urban
environment have not been quantified before. The results of our estimates suggest that NPF has a stronger influence on
the sub-3 nm particle population than traffic at the urban background site, especially during the daytime. At the street
canyon site, NPF and traffic contribute to sub-3 nm concentrations rather equally. This indicates that traffic is an important
source of sub-3 nm particles, but it does not solely dominate the sub-3 nm particle population at either of the studied sites
in Helsinki during the daytime in spring. However, in our estimation, we considered only the process of formation or
emission of sub-3 nm particles, and thus we did not account for the origin of NPF precursors. SA, as well as other low-
volatile compounds, can be emitted by traffic and then participate in the formation of sub-3 nm particles. Furthermore,
one should note that this estimation is performed only with a limited data set, and therefore it may not provide a full
picture of the contributions of NPF and traffic to sub-3 nm particles in Helsinki. The relations between emissions of
particles and NOx from traffic and between NPF and SA are expected to vary seasonally or as a function of temperature
(Gidhagen et al., 2005; Nieminen et al., 2014) and, consequently, the parameters derived in this study are not expected to
be valid through the year in Helsinki, even less in other locations. For instance, NPF events are frequently observed in
Finland in spring and autumn but very seldom in winter (Hussein et al., 2008), and particle emissions from traffic are
expected to be higher during colder temperatures in winter (Gidhagen et al., 2005). Since this study was conducted in
spring, the role of NPF events as a sub-3 nm particles source would probably be much smaller in winter.
Future studies should focus on different low-volatile compounds in an urban environment and investigate their influence
on NPF. Additionally, analyzing the relative influence of different processes on the sub-3 nm particle population based
on long-term measurements would be beneficial. In the future, the method developed in this study can also be applied to
estimate the contribution of traffic to particle number concentrations in other urban enviroments. This knowledge can
contribute to a better understanding of the effects of traffic on air quality and human health.



**Appendix**
Table A1. Working time of instruments used during the campaign.

| Instrument | Model | Working period | Working time [h] |
|---|---|---|---|
| | | Street canyon | |
| Particle Size Magnifier (PSM ) fixed mode | Airmodus A11 nCNC | 27.04.2018 13:00 – 02.05.2018 08:00<br>04.05.2018 09:00 – 07.05.2018 15:00<br>08.05.2018 10:00 – 09.05.2018 14:00<br>10.05.2018 10:00 – 10.05.2018 14:00<br>11.05.2018 01:00 – 11.05.2018 17:00 | 246 |
| PSM step mode | Airmodus A11 nCNC | 27.04.2018 00:00 – 15.05.2018 12:00<br>15.05.2018 21:00 – 01.05.2018 00:00 | 833 |
| Ultrafine Condensation Particle Counter (UCPC) | TSI 3776 | 27.04.2018 14:00 – 30.04.2018 07:00<br>03.05.2018 19:00 – 06.05.2018 09:00<br>08.05.2018 11:00 – 18.05.2018 16:00<br>21.05.2018 18:00 – 21.05.2018 22:00<br>24.05.2018 10:00 – 24.05.2018 12:00<br>24.05.2018 16:00 – 31.05.2018 10:00 | 548 |
| Condensation Particle Counter A20 (CPC) | Airmodus A20 | 27.04.2018 14:00 – 17.05.2018 15:00<br>17.05.2018 23:00 – 22.05.2018 10:00<br>24.05.2018 10:00 – 25.05.2018 14:00<br>26.05.2018 00:00 – 31.05.2018 10:00 | 750 |
| Differential Mobility Particle Sizer (DMPS) | Vienna-type DMA coupled with Airmodus A20 DMA | 27.04.2018 00:00 – 31.05.2018 23:00 | 840 |
| Atmospheric Pressure Interface Time-Of-Flight Mass Spectrometer with the Chemical Ionization (CI-APi-TOF) | TOFWERK AG | 02.05.2018 00:00 – 03.05.2018 13:00<br>04.05.2018 15:00 – 08.05.2018 09:00<br>08.05.2018 13:00 – 08.05.2018 15:00<br>08.05.2018 18:00 – 10.05.2018 09:00<br>10.05.2018 12:00 – 16.05.2018 09:00<br>16.05.2018 12:00 – 22.05.2018 10:00<br>22.05.2018 12:00 – 24.05.2018 07:00 | 501 |
| $SO_2$ analyzer | 43i-TLE | 09.05.2018 12:00 – 10.05.2018 17:00<br>11.05.2018 09:00 – 11.05.2018 18:00<br>14.05.2018 09:00 – 31.05.2018 12:00 | 452 |
| $CO_2$ analyzer | LI-COR LI-7000 | 27.04.2018 09:00 – 24.05.2018 13:00<br>24.05.2018 16:00 – 29.05.2018 19:00 | 771 |
| $CO_2$ analyzer (Tut) | | 17.05.2018 14:00 – 19.05.2018 00:00<br>21.05.2018 17:00 – 21.05.2018 20:00<br>24.05.2018 10:00 – 24.05.2018 11:00 | 90 |





| | | 25.05.2018 12:00 – 25.05.2018 13:00 | |
| | | 28.05.2018 10:00 – 28.05.2018 14:00 | |
| | | 28.05.2018 16:00 – 29.05.2018 08:00 | |
| | | 30.05.2018 11:00 – 31.05.2018 10:00 | |
| Background | | | |
| PSM fixed mode | Airmodus A11 nCNC | 01.05.2018 10:00 – 10.05.2018 04:00 | 766 |
| | | 11.05.2018 08:00 – 05.06.2018 10:00 | |
| PSM scanning mode | Airmodus A11 nCNC | 04.05.2018 00:00 – 21.05.2018 11:00 | 671 |
| | | 21.05.2018 22:00 – 30.05.2018 05:00 | |
| | | 02.06.2018 22:00 – 05.06.2018 00:00 | |
| UCPC | TSI 3776 | 03.05.2018 17:00 – 10.05.2018 04:00 | 670 |
| | | 11.05.2018 08:00 – 21.05.2018 16:00 | |
| | | 25.05.2018 10:00 – 05.06.2018 10:00 | |
| CPC | Airmodus A20 | 03.05.2018 11:00 – 10.05.2018 04:00 | 728 |
| | | 11.05.2018 08:00 – 20.05.2018 00:00 | |
| | | 21.05.2018 14:00 – 05.06.2018 10:00 | |
| Twin Differential Mobility Particle Sizer (Twin-DMPS) | | 01.05.2018 01:00 – 07.05.2018 14:00 | 821 |
| | | 08.05.2018 09:00 – 05.06.2018 00:00 | |
| Neutral and Air Ion Spectrometer (NAIS) | Airel Ltd | 26.04.2018 13:00 – 17.05.2018 02:00 | 932 |
| | | 18.05.2018 21:00 – 06.06.2018 02:00 | |
| CI-APi-TOF | TOFWERK AG | 02.05.2018 15:00 – 04.06.2018 00:00 | 776 |
| $SO_2$ analyzer | | 27.04.2018 01:00 – 07.05.2018 12:00 | 937 |
| | | 08.05.2018 11:00 – 06.06.2018 00:00 | |
| $CO_2$ analyzer | | 03.05.2018 18:00 – 14.05.2018 01:00 | 248 |

Table A2. Overview of additional variables used in this study measured at the background station and the street canyon
site.

| Variable [unit] | Instrument / model | Height (m) |
|---|---|---|
| Background station | | |
| NO, $NO_x$ [ppb] | chemiluminescence analyzer / Horiba APNA 370 | 4 |
| $O_3$ | UV-absoption / Teledyne Instruments API 400E | 4 |
| $SO_2$ [ppb] | UV fluorescence analyzer / Thermo Fisher Scientific TEI 43iTLE | 4 |
| Relative humidity [%] | Vaisala HMP243 | 29 |
| Air temperature [°C] | Pentronic Pt100 | 4 |
| Wind direction [°] | 2D ultrasonic anemometer/ Thies Clima 2.1x | 32 |
| Wind speed [m/s] | 2D ultrasonic anemometer/ Thies Clima 2.1x | 32 |
| Global radiation [W/m$^2$] | Kipp and Zonen CNR1 | 32 |





| Black carbon | Multi Angle Absorption Photometer (MAAP), Thermo Scientific, Model 5012 | 4 |
|---|---|---|
| Ion size distribution [cm⁻³] | neutral cluster and air ion spectrometer (NAIS) | 1.5 |
| Street canyon | | |
| NO, NOₓ [µg/m3] | chemiluminescence analyzer / Horiba APNA 370 | 4 |
| O₃ [µg/m3] | UV fluorescence analyzer / Horiba APOA-370 | 4 |
| Relative humidity [%] | Vaisala WXT 536 | 4 |
| Air temperature [°C] | Vaisala WXT 536 | 4 |
| Wind direction [°] | Vaisala WXT 536 | 4 |
| Wind speed [m/s] | Vaisala WXT 536 | 4 |
| Black carbon | Optical analyzer / MAAP 5012 | 4 |

**Calculating the relative contribution of traffic and NPF to sub-3 nm particle population**
Based on bivariate fittings on the common logarithms of sub-3 nm particles and SA when $NO_x$ concentration was low at
the street canyon site (Fig. 9a), we determined Eq. A1 estimating the concentration of sub-3 nm particles formed during
NPF ($[N_{1-3}]_{SA}$) at the street canyon site. The same analysis conducted for the concentrations at the background station
(Fig. 9b) resulted in Eq. A2. Correlation between common logarithms of sub-3 nm particles and $NO_x$, when SA
concentration was low, at the street canyon and background station (Fig. 9c-d) was used to determine Eq. A3 and A4,
respectively. Equations A3 and A4 estimate the concentration of sub-3 nm particles emitted from traffic ($[N_{1-3}]_{NO_x}$).

$$[N_{1-3}]_{SA} = 10^{-4.04} \cdot [SA]^{1.25} \qquad \text{(street canyon)} \qquad \text{(A1)}$$

where $[N_{1-3}]_{SA}$ is an estimated concentration of sub-3 nm particles formed during NPF, and [SA] is a SA concentration.

$$[N_{1-3}]_{SA} = 10^{-2.86} \cdot [SA]^{0.99} \qquad \text{(background)} \qquad \text{(A2)}$$

$$[N_{1-3}]_{NO_x} = 10^{-2.45} \cdot [NO_x]^{1.40} \qquad \text{(street canyon)} \qquad \text{(A3)}$$

where $[N_{1-3}]_{NO_x}$ an estimated concentration of sub-3 nm particles emitted from traffic, and [NOₓ] is a NOₓ concentration.

$$[N_{1-3}]_{NO_x} = 10^{0.59} \cdot [NO_x]^{0.64} \qquad \text{(background)} \qquad \text{(A4)}$$

Based on Eq. A1-A4, the relative contribution of traffic and NPF at each site was computed. To calculate the relative
contribution of traffic ($x_{[N_{1-3}]NO_x}$), the estimated concentration of sub-3 nm particles emitted by traffic was divided by
the sum of estimated concentrations of sub-3 nm particles emitted by traffic and formed during NPF (Eq. A5). Similarly,
the relative contribution of NPF ($x_{[N_{1-3}]SA}$) was computed by dividing the estimated concentration of sub-3 nm particles
formed during NPF by the sum of estimated concentrations of sub-3 nm particles emitted by traffic and formed during
NPF (Eq. A6). The relative contribution of each source was calculated for the street canyon and the background station.

$$x_{[N_{1-3}]NO_x} = \frac{[N_{1-3}]_{NO_x}}{[N_{1-3}]_{SA} + [N_{1-3}]_{NO_x}} \cdot 100\% \qquad \text{(A5)}$$



where $x_{[N_{1-3}]_{NO_x}}$ is a relative contribution of traffic, $[N_{1-3}]_{NO_x}$ is an estimated concentration of sub-3 nm particles emitted by traffic, and $[N_{1-3}]_{SA}$ is an estimated concentration of sub-3 nm particles formed during NPF.

$$x_{[N_{1-3}]_{SA}} = \frac{[N_{1-3}]_{SA}}{[N_{1-3}]_{SA} + [N_{1-3}]_{NO_x}} \cdot 100\% \tag{A6}$$

where $x_{[N_{1-3}]_{SA}}$ is a relative contribution of NPF, $[N_{1-3}]_{NO_x}$ is an estimated concentration of sub-3 nm particles emitted by traffic, and $[N_{1-3}]_{SA}$ is an estimated concentration of sub-3 nm particles formed during NPF.

**Data availability**

Data will be published in an open data repository. DMPS, BC, $O_3$, meteorological data measured at the background station are available at the SmartSMEAR data repository (https://avaa.tdata.fi/web/smart).

**Author contribution**

The main ideas were formulated by TP, PP, JKo, HK, JVN and the results were interpreted by MOk, JKo, HK, PP, and TR. HK, MA, KT, HL, LS prepared measurement methodology and OG, HK, MOl, RB, HP, MA, HL, and LS contributed to data collection. MOk, HK, KT, RB performed the data analysis. Instruments were calibrated by MOl, JKa, YJT, and RB. MS, TP, TR, HT, JVN coordinated project while JKo, OG, PP, JKa, and HT supervised it. MS, TP, HT, TR made a funding acquisition. MOk visualized data and prepared the manuscript with contributions from other authors. All the authors reviewed the manuscript.

**Competing interests**

The authors declare that they have no conflict of interest.

**Acknowledgments.**

This research was supported by the Regional innovations and experimentations funds AIKO (project HAQT, AIKO014), Business Finland (CITYZER project, Tekes nro: 3021/31/2015 and 2883/31/2015), Pegasor Oy and HSY, Academy of Finland (grant nos 273010, 307331, 310626, 311932, 316114, 318940, 1325656 & 326437), Healthy Outdoor Premises for Everyone (HOPE), Urban Innovation Actions, Regional development funds, University of Helsinki doctoral programme (ATM-DP) and Faculty of Science 3-year grant (75284132), Tampere University of Technology graduate school, and ERA-PLANET project SMURBS (Grant Agreement 689443) under the EU Horizon 2020 Framework.

We would like to thank the people who took care of instruments and measurements at the SMEAR III (Pekka Rantala, Erkki Siivola, Pasi Aalto, Petri Keronen, Frans Korhonen, Tiia Laurila, Lauriane Quéléver, Tuuli Lehmusjärvi, Deniz Kemppainen) and the HSY Mäkelänkatu site (Anssi Julkunen, Anders Svens, Taneli Mäkelä, Tommi Wallenius, Anu Kousa). In addition, we thank Jiali Shen and Xucheng He for help with the calibration of CI-APi-TOF and Lei Yao for useful discussions.

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

Methods for determining particle size distribution and growth rates between 1 and 3 nm using the Particle Size
Magnifier, Boreal Environ. Res., 19(September), 215–236, 2014.
Maher, B. A., Ahmed, I. A. M., Karloukovski, V., MacLaren, D. A., Foulds, P. G., Allsop, D., Mann, D. M. A., Torres-
Jardón, R. and Calderon-Garciduenas, L.: Magnetite pollution nanoparticles in the human brain, Proc. Natl. Acad. Sci.
U. S. A., 113(39), 10797–10801, doi:10.1073/pnas.1605941113, 2016.
Mårtensson, E. M., Nilsson, E. D., Buzorius, G. and Johansson, C.: Eddy covariance measurements and
parameterisation of traffic related particle emissions in an urban environment, Atmos. Chem. Phys., 6(3), 769–785,
doi:10.5194/acp-6-769-2006, 2006.
Mauldin, R. L., Tanner, D. J., Heath, J. A., Huebert, B. J. and Eisele, F. L.: Observations of H2SO4 and MSA during





PEM-Tropics-A, J. Geophys. Res. Atmos., 104(D5), 5801–5816, doi:10.1029/98JD02612@10.1002/(ISSN)2169-
8996.PAMTA1, 1999.
Mylläri, F., Asmi, E., Anttila, T., Saukko, E., Vakkari, V., Pirjola, L., Hillamo, R., Laurila, T., Häyrinen, A.,
Rautiainen, J., Lihavainen, H., O'Connor, E., Niemelä, V., Keskinen, J., Dal Maso, M. and Rönkkö, T.: New particle
formation in the fresh flue-gas plume from a coal-fired power plant: effect of flue-gas cleaning, Atmos. Chem. Phys.,
16(11), 7485–7496, doi:10.5194/acp-16-7485-2016, 2016.
Nieminen, T., Asmi, A., Dal maso, M., Aalto, P. P., Keronen, P., Petäjä, T., Kulmala, M. and Kerminen, V.: Trends in
atmospheric new-particle formation: 16 years of observations in a boreal-forest environment. [online] Available from:
http://www.arl. (Accessed 11 January 2021), 2014.
Olin, M., Kuuluvainen, H., Aurela, M., Kalliokoski, J., Kuittinen, N., Isotalo, M., Timonen, H. J., Niemi, J. V., Rönkkö,
T. and Dal Maso, M.: Traffic-originated nanocluster emission exceeds H2SO4-driven photochemical new particle
formation in an urban area, Atmos. Chem. Phys., 20(1), 1–13, doi:10.5194/acp-20-1-2020, 2020.
Pirjola, L., Paasonen, P., Pfeiffer, D., Hussein, T., Hämeri, K., Koskentalo, T., Virtanen, A., Rönkkö, T., Keskinen, J.,
Pakkanen, T. A. and Hillamo, R. E.: Dispersion of particles and trace gases nearby a city highway: Mobile laboratory
measurements in Finland, Atmos. Environ., 40(5), 867–879, doi:10.1016/j.atmosenv.2005.10.018, 2006.
Ramanathan, V. and Feng, Y.: Air pollution, greenhouse gases and climate change: Global and regional perspectives,
Atmos. Environ., 43(1), 37–50, doi:10.1016/j.atmosenv.2008.09.063, 2009.
Ripamonti, G., Järvi, L., Mølgaard, B., Hussein, T., Nordbo, A. and Hämeri, K.: The effect of local sources on aerosol
particle number size distribution, concentrations and fluxes in Helsinki, Finland, Tellus, Ser. B Chem. Phys. Meteorol.,
65(1), 19786, doi:10.3402/tellusb.v65i0.19786, 2013.
Rönkkö, T. and Timonen, H.: Overview of Sources and Characteristics of Nanoparticles in Urban Traffic-Influenced
Areas, J. Alzheimer's Dis., 72(1), 15–28, doi:10.3233/JAD-190170, 2019.
Rönkkö, T., Kuuluvainen, H., Karjalainen, P., Keskinen, J., Hillamo, R., Niemi, J. V., Pirjola, L., Timonen, H. J.,
Saarikoski, S., Saukko, E., Järvinen, A., Silvennoinen, H., Rostedt, A., Olin, M., Yli-Ojanperä, J., Nousiainen, P.,
Kousa, A. and Dal Maso, M.: Traffic is a major source of atmospheric nanocluster aerosol, Proc. Natl. Acad. Sci. U. S.
A., 114(29), 7549–7554, doi:10.1073/pnas.1700830114, 2017.
Rose, C., Zha, Q., Dada, L., Yan, C., Lehtipalo, K., Junninen, H., Mazon, S. B., Jokinen, T., Sarnela, N., Sipilä, M.,
Petäjä, T., Kerminen, V. M., Bianchi, F. and Kulmala, M.: Observations of biogenic ion-induced cluster formation in
the atmosphere, Sci. Adv., 4(4), 1–11, doi:10.1126/sciadv.aar5218, 2018.
Rosenfeld, D., Lohmann, U., Raga, G. B., O'Dowd, C. D., Kulmala, M., Fuzzi, S., Reissell, A. and Andreae, M. O.:
Flood or drought: How do aerosols affect precipitation?, Science (80-. )., 321(5894), 1309–1313,
doi:10.1126/science.1160606, 2008.
Salma, I., Borsós, T., Weidinger, T., Aalto, P., Hussein, T., Dal Maso, M. and Kulmala, M.: Production, growth and
properties of ultrafine atmospheric aerosol particles in an urban environment, Atmos. Chem. Phys., 11(3), 1339–1353,
doi:10.5194/acp-11-1339-2011, 2011.





Seinfeld, J. H. and Pandis, S. N.: Atmospheric Chemistry and Physics: From Air Pollution to Climate Change., 2016.
Sipila, M., Berndt, T., Petaja, T., Brus, D., Vanhanen, J., Stratmann, F., Patokoski, J., Mauldin, R. L., Hyvärinen, A. P.,
Lihavainen, H. and Kulmala, M.: The role of sulfuric acid in atmospheric nucleation, Science (80-. )., 327(5970), 1243–
1246, doi:10.1126/science.1180315, 2010.
Tauber, C., Brilke, S., Wlasits, P. J., Bauer, P. S., Köberl, G., Steiner, G. and Winkler, P. M.: Humidity effects on the
detection of soluble and insoluble nanoparticles in butanol operated condensation particle counters, Atmos. Meas.
Tech., 12(7), 3659–3671, doi:10.5194/amt-12-3659-2019, 2019.
Tian, L., Shang, Y., Chen, R., Bai, R., Chen, C., Inthavong, K. and Tu, J.: Correlation of regional deposition dosage for
inhaled nanoparticles in human and rat olfactory, Part. Fibre Toxicol., 16(1), 6, doi:10.1186/s12989-019-0290-8, 2019.
Vanhanen, J., Mikkilä, J., Lehtipalo, K., Sipilä, M., Manninen, H. E., Siivola, E., Petäjä, T. and Kulmala, M.: Particle
Size Magnifier for Nano-CN Detection, Aerosol Sci. Technol., 45(4), 533–542, doi:10.1080/02786826.2010.547889,
714    2011.

Viggiano, A. A., Seeley, J. V., Mundis, P. L., Williamson, J. S. and Morris, R. A.: Rate Constants for the Reactions of
XO 3 - (H 2 O) n (X = C, HC, and N) and NO 3 - (HNO 3 ) n with H 2 SO 4 : Implications for Atmospheric
Detection of H 2 SO 4 , J. Phys. Chem. A, 101(44), 8275–8278, doi:10.1021/jp971768h, 1997.
Wang, Z. B., Hu, M., Yue, D. L., Zheng, J., Zhang, R. Y., Wiedensohler, A., Wu, Z. J., Nieminen, T. and Boy, M.:
Evaluation on the role of sulfuric acid in the mechanisms of new particle formation for Beijing case, Atmos. Chem.
Phys., 11(24), 12663–12671, doi:10.5194/acp-11-12663-2011, 2011.
Wiedensohler, A., Birmili, W., Nowak, A., Sonntag, A., Weinhold, K., Merkel, M., Wehner, B., Tuch, T., Pfeifer, S.,
Fiebig, M., Fjäraa, A. M., Asmi, E., Sellegri, K., Depuy, R., Venzac, H., Villani, P., Laj, P., Aalto, P., Ogren, J. A.,
Swietlicki, E., Williams, P., Roldin, P., Quincey, P., Hüglin, C., Fierz-Schmidhauser, R., Gysel, M., Weingartner, E.,
Riccobono, F., Santos, S., Grüning, C., Faloon, K., Beddows, D., Harrison, R., Monahan, C., Jennings, S. G.,
O&apos;Dowd, C. D., Marinoni, A., Horn, H.-G., Keck, L., Jiang, J., Scheckman, J., McMurry, P. H., Deng, Z.,
Zhao, C. S., Moerman, M., Henzing, B., de Leeuw, G., Löschau, G. and Bastian, S.: Mobility particle size
spectrometers: harmonization of technical standards and data structure to facilitate high quality long-term observations
of atmospheric particle number size distributions, Atmos. Meas. Tech., 5(3), 657–685, doi:10.5194/amt-5-657-2012,
729    2012.

Williamson, J. H.: Least-squares fitting o f a straight line, Can. J. Phys., 46(16), 1845–1847, doi:10.1139/p68-523,
731    1968.

Winkler, P. M., Steiner, G., Vrtala, A., Vehkamäki, H., Noppel, M., Lehtinen, K. E. J., Reischl, G. P., Wagner, P. E.
and Kulmala, M.: Heterogeneous nucleation experiments bridging the scale from molecular ion clusters to
nanoparticles, Science (80-. )., 319(5868), 1374–1377, doi:10.1126/science.1149034, 2008.
Wlasits, P. J., Stolzenburg, D., Tauber, C., Brilke, S., Schmitt, S. H., Winkler, P. M. and Wimmer, D.: Counting on
chemistry: laboratory evaluation of seed-material-dependent detection efficiencies of ultrafine condensation particle
counters, Atmos. Meas. Tech., 13(7), 3787–3798, doi:10.5194/amt-13-3787-2020, 2020.



Wonaschütz, A., Demattio, A., Wagner, R., Burkart, J., Zíková, N., Vodička, P., Ludwig, W., Steiner, G., Schwarz, J.
and Hitzenberger, R.: Seasonality of new particle formation in Vienna, Austria - Influence of air mass origin and
aerosol chemical composition, Atmos. Environ., 118, 118–126, doi:10.1016/j.atmosenv.2015.07.035, 2015.
Xiao, S., Wang, M. Y., Yao, L., Kulmala, M., Zhou, B., Yang, X., Chen, J. M., Wang, D. F., Fu, Q. Y., Worsnop, D. R.
and Wang, L.: Strong atmospheric new particle formation in winter in urban Shanghai, China, Atmos. Chem. Phys.,
15(4), 1769–1781, doi:10.5194/acp-15-1769-2015, 2015.
Yao, L., Garmash, O., Bianchi, F., Zheng, J., Yan, C., Kontkanen, J., Junninen, H., Mazon, S. B., Ehn, M., Paasonen,
P., Sipilä, M., Wang, M., Wang, X., Xiao, S., Chen, H., Lu, Y., Zhang, B., Wang, D., Fu, Q., Geng, F., Li, L., Wang,
H., Qiao, L., Yang, X., Chen, J., Kerminen, V. M., Petäjä, T., Worsnop, D. R., Kulmala, M. and Wang, L.: Atmospheric
new particle formation from sulfuric acid and amines in a Chinese megacity, Science (80-. )., 361(6399), 278–281,
doi:10.1126/science.aao4839, 2018.
York, D.: Least-squares fitting of a straight line, Can. J. Phys., 44, 1079–1086, 1966.
Zhou, Y., Dada, L., Liu, Y. Y., Fu, Y., Kangasluoma, J., Chan, T., Yan, C., Chu, B., Daellenbach, K. R., Bianchi, F.,
Kokkonen, T. V., Liu, Y. Y., Kujansuu, J., Kerminen, V. M., Petäjä, T., Wang, L., Jiang, J., Kulmala, M., Xiao, S.,
Wang, M. Y., Yao, L., Kulmala, M., Zhou, B., Yang, X., Chen, J. M., Wang, D. F., Fu, Q. Y., Worsnop, D. R. and
Wang, L.: Variation of size-segregated particle number concentrations in wintertime Beijing, Atmos. Chem. Phys.,
20(2), 1201–1216, doi:10.5194/acp-20-1201-2020, 2020.
Zhu, Y., Hinds, W. C., Kim, S. and Sioutas, C.: Concentration and size distribution of ultrafine particles near a major
highway, J. Air Waste Manag. Assoc., 52(9), 1032–1042, doi:10.1080/10473289.2002.10470842, 2002.