# Peer review of "Measurement report: The influence of traffic and new particle"

_Atmospheric Chemistry and Physics, 2020_

## Author Response (AR1)

We thank the reviewers for their valuable comments and suggestions that improved our manuscript. Here, we present answer for each of their comments. Reviewers' comments are quoted as a **bold** text, while the text from the manuscript is marked with *italics* and the changes in the text are highlighted in grey.

**Reply to anonymous referee #1**

**The manuscript by Okuljar et al builds on the legacy of urban particle and gas measurements in the city of Helsinki, the novelty being 1-3 nm particle measurements with simultaenous measurements of primarily new particle formation (NPF) vs. primarily traffic gas phase tracers, sulphuric acid (SA), and NOx, respectively. This combination of measurements, especially under the aspect of two collocated sites, one traffic the other urban background, is unique and yields insight into important processes that in turn are critical for future measures concerning air quality and human health. Thus, I recommend to consider this work for publication.**

**In my opinion, this work and the presented partitioning approach could further increase their impact by a more quantitative presentation of the key results together with the uncertainties and how the latter could challenge the most important findings here, e.g. that during daytime the direct traffic source of 1-3 nm particles can be minor (vs. the NPF source). It would be much appreciated if you could provide more detailed suggestions on how to adopt your reggression approach (or the findings from it in your study) to other sites or other tracers to help adressing the knowledge gaps and uncertainties present in current generation air quality models. E.g. as of now the reader does not know whether your findings (e.g. the one summarized above) is in line or challenges current approaches on estimating total N of very small particles in urban areas. I provide some detailed comments below, and I'm looking forward to reading your work in the future.**

We thank the reviewer for the encouraging comments. We discuss in the answers below the sources of uncertainty mentioned by the reviewer. We also want to point out that we cannot generalize our results too much due to the limitations of our data set: we had only a short measurement period with few overlapping days between the two measurements sites, and only a limited number of NPF events occurring on these days. Therefore, a study utilizing longer-term data set is needed to be able to confirm our results and present them more quantitatively. This is discussed in the second to last paragraph in the conclusions.

In addition, we want to clarify that our approach to estimate contributions of NPF and traffic based on sulfuric acid and $NO_x$ should not be directly applied to other measurement sites. This is because, in different environments, there can be different sources of $NO_x$ and sulfuric acid. We make this now clear in the manuscript by modifying the following sentences in the abstract and in the conclusions:

Line 27: *In the future, the contribution of traffic to particle number concentrations in different urban environments can be estimated with a similar approach, but determining the relationships between the gas and particle concentrations from observations needs to be conducted with longer data sets from different urban environments.*

Line 513: *In the future, the approach to estimate the contribution of NPF and traffic to particle number concentrations based on trace-gas concentration may be applied in other urban environments, which can improve the understanding of the effects of traffic on urban air quality. However, the variables, parameters and functions describing the traffic and NPF contributions may vary between sites and season, which leaves the determination of more universal functions to be the scope of future studies, with longer time series from different urban areas.*

Previous papers estimating the total concentration of very small particles in urban areas usually do not include particles smaller than 3 nm. Olin et al. (2020) is the only paper estimating sub-3 nm particle

concentration in urban areas. A detailed comparison of our finding with Olin et al. (Olin et al., 2020) is included in the answer for the following comment.

**General method:**

**In the introductions you report that also SA has a indirect and direct traffic source (L53&54), however in L209 you describe your usage of SA as a NPF tracer in your correlation/partitioning method (vs. NOx being the traffic tracer). Only much later in your case study description we learn that based on your observations SA seems to be much less correlated to traffic than NOx and sub 3 nm (street site). You acknowledge this uncertainty in L417 and state that it could lead to an underestimation of the traffic contribution to sub 3 nm N in your method. Could you investigate this error in a more quantitative way, how strong is the correlation between SA and NOx and does it vary as a function of traffic rate? By how much do you think you might underestimate the traffic 3 nm particle source due to this effect and does this underestimation vary under different conditions? Could this uncertainty challenge your findings of only small traffic sources of sub 3 nm N during NPF?**

Thanks for the good comment. In our estimation, the contribution by traffic should include other direct and indirect emissions of sub-3 nm particles except for the production of sub-3 nm particles by SA that originates from the photochemical oxidation of traffic-emitted $SO_2$ outside the immediate vicinity to the emission.

It is a good suggestion to investigate the correlation between SA and NOx as a function of traffic rate. Unfortunately, traffic rate data is not available for the measurement period. Instead, we estimated the possibility of traffic emissions being interpreted as NPF particles by investigating the relations between SA and NOx concentrations and between NOx and $N_{1-3}$ during daytime (6-20 hours) (Fig. R1). We chose to use daytime data only since both traffic and photochemical SA production have similar diurnal cycles (minima during nights and maxima during daytime). Figures R1a and R1c show that sulfuric acid concentration is not closely connected to NOx concentrations even in the street canyon but more to atmospheric oxidation chemistry. The coefficient of determination ($R^2$) shows that the variance in $NO_x$ concentration explains only 7 % of the variance the sulfuric acid concentration even in the immediate vicinity to the emissions. When we investigated the relation between SA and NOx in different bins of global radiation, we observed that in none of the three radiation bins (radiation < 20 $W/m^2$, 20-300 $W/m^2$ and >300 $W/m^2$) at the street canyon site, the coefficient of determination exceeded 0.10 (Table R1). Thus, we are convinced to state that the association between SA and $N_{1-3}$ (in Figures 9a and 9c in the manuscript) can be interpreted as the influence of atmospheric NPF on $N_{1-3}$ concentration.

[Figure]

Figure R1. Relations between SA and $NO_x$ concentrations (a,c) and sub-3 nm particles and $NO_x$ concentrations (b,d) at the street canyon site (a,b) and the background station (c,d) calculated for the daytime (6-20) data with ten minutes resolution. The data points are colored with radiation level observed at the background station.

Table R1. Coefficient of determination ($R^2$) between SA and NOx concentrations during daytime (10 minutes averaged data between 6 and 20 hours) at the study sites in different ranges of global radiation. The sample size is given in parenthesis, and the correlations not significant at $p<0.05$ level are shown in italics.

| | Low radiation (<20 W/m$^2$) | Medium radiation (20-300 W/m$^2$) | High radiation (>300 W/m$^2$) |
|---|---|---|---|
| Street canyon | 0.018 (525) | 0.067 (409) | *0 (371)* |
| Background | 0.035 (21) | 0.014 (740) | 0.117 (1536) |

However, Olin et al. (2020) estimated that 68% of the SA originates from traffic at the same street canyon site on a typical weekday at noon in May 2017, a year before the measurements reported here. This finding is not in line with our results presented in Fig R1 and table R1. We investigated the data from May 2017 and 2018, and the correlation between SA and NOx concentrations is indeed much higher in 2017 than in 2018. In both data sets, the $N_{1-3}$ concentration is tightly connected to NOx concentration. We do not know why the two data sets differ. However, investigating this further is not within the scope of this manuscript, but it underlines the need for continuous and long-term observations of these variables to better understand and parameterize the roles of traffic and NPF on urban aerosol. We also leave the detailed comparison between different years at the street canyon site for future studies since this study aims to compare the street canyon data with the background data. We add text as indicated below to reflect the differences between 2017 and 2018 and the above conclusions. Also, the edits to the very last paragraph of the Conclusions, described in the reply to the first question above, are related to the discussion here.

We add Fig R1 and Table R1 to the supplementary material of the manuscript and make the following changes in the text:

Lines 218: *NOx concentration was used as a traffic marker (Olin et al., 2020) while SA concentration was used as an NPF marker (Sipila et al., 2010). We separately investigated the relation between SA and NO$_x$ concentrations in the street canyon, in order to justify the use of SA as a tracer for NPF even though emissions of SA from traffic have previously been reported (Arnold et al., 2012; Olin et al., 2020).*

Line 407: *We used the compounds that correlate best with sub-3 nm particles at each site, SA and NO$_x$, as tracers for NPF and traffic emissions, respectively. Using SA concentration as a tracer for NPF is based on the assumption that traffic is not a major source of SA. We justify and discuss this assumption at the end of this section.*

Line 433: *We estimate that during the daytime (6:00-20:00), a similar fraction of sub-3 nm particles originates from traffic (53%) and NPF (47%) at the street canyon site.*

Line 439: *When discussing the estimated relative contribution of traffic and NPF to the sub-3 nm population, we should keep in mind that the conducted analysis does not consider the origin of SA. Traffic can directly or indirectly emit SA, thus traffic may influence SA concentration used for estimating sub-3 nm particles formed during NPF. This could cause an underestimation of the relative contribution of traffic to the sub-3 nm population. In order to estimate the significance of this underestimation, we investigated the relations between SA and NOx concentrations and between NOx and N1-3. We found that even in the street canyon site, the variance in NOx concentration explained at maximum only 7 % of the daytime variance in SA concentration (Figure S18 and Table S3), which justifies the use of SA as NPF tracer. However, Olin et al. (2020) estimated that during typical workday noon in May 2017, one year prior to the observations described here, 68% of SA concentration at the same street canyon site originated from traffic, which is not in line with our observations for May 2018. The reason for this discrepancy is not known. Additionally, Olin et al. (2020) estimated that during May 2017, at typical workday noontime at the same street canyon site, the contribution of traffic to sub-3 nm particles was approximately 85%, which is clearly higher than our estimate. The difference between our results and the ones presented by Olin et al. (2020) could be partly caused by the difference between the influence of traffic on SA concentration between the two measurement campaigns, described above. Additionally, one needs to keep in mind that Olin et al. (2020) calculated the traffic contribution to the sub-3 nm particles for a typical workday, while most of our data (57.8%) from the street canyon site was collected at the time free from work. Overall, the differences between our results and those by Olin et al. (2020) can indicate that even at the same site and at the same time of the year, the emissions and formation of sub-3 nm particles may follow different mechanisms.*

Line 457: *Generally, one should note that the estimates presented here are based on only a limited data set, and therefore it is not expected to provide a complete picture of the contributions of NPF and traffic to sub-3 nm particles in Helsinki. The relations between emissions of particles and NOx from traffic and between NPF and SA are expected to vary seasonally or as a function of temperature (Gidhagen et al., 2005; Nieminen et al., 2014) and, consequently, the parameters derived in this study are not expected to be valid through the year in Helsinki, even less in other locations. For instance, NPF events are frequently observed in Finland in spring and autumn but very seldom in winter (Hussein et al., 2008), and particle emissions from traffic are expected to be higher during colder temperatures in winter (Gidhagen et al., 2005). In winter, the role of NPF events as a sub-3 nm particles source would probably be much smaller than what we estimated in this study conducted in spring.*

Line 505: *However, in our estimation, we did not account for the origin of NPF precursors. SA and other low-volatile compounds can be emitted by traffic and then participate in the formation of sub-3 nm particles. For our data set, we showed that the SA emissions from traffic are clearly lower to photochemical SA formation, justifying our estimates of distinguishing traffic and NPF particles based on SA and NOx concentrations. This might not be as simple in other cases, since Olin et al. (2020) showed that at the same site in the previous spring, the traffic emissions played much more significant role in SA concentrations.*

**You use the fitting coefficients for your partitioning model from correlations based on arbitrarily chosen background conditions, i.e. times when either NOx was low (for the sub 3 nm N vs. SA regression), or when SA was low (for the sub 3 nm vs NOx regression). If then applied to all data, how do you deal with measurements from times when NOx, or SA, were not low/or high. The regression coefficients might vary in the intermediate regimes (based on Fig. 9 and SI figure).**

The background conditions that we are using for the estimation analysis (low concentration of $NO_x$ for sub-3 nm particles vs SA analysis and low concentration of SA for sub-3 nm particles vs $NO_x$ analysis) are needed to prevent overestimation of the total sub-3 nm particles concentration. The estimation analysis is based on the assumption that we can separate sub-3 nm particles emitted by traffic ($[N_{1-3}]_{NOx}$) and formed during NPF processes ($[N_{1-3}]_{SA}$). The sum of these two components should represent the total concentration of sub-3 nm particles at the studied location. For example, if we used data from times when the $NO_x$ is not low, for fitting SA vs sub-3 nm particle concentrations, we would likely include some particles emitted by traffic into the $[N_{1-3}]_{SA}$ component. That would lead to overestimation of total sub3-nm particles concentration since some particles would be included in both components ($[N_{1-3}]_{SA}$ and $[N_{1-3}]_{NOx}$). It would also make it impossible to estimate the relative contribution of traffic and NPF to the sub-3 nm particle population.

If we look at the correlation between sub-3 nm particles concentration and NOx at the background station (Fig 9 d, Table S2), the slope and the correlation coefficient stays constant for $\log_{10}(SA)<6.5$. For higher SA concentration, the slope increases and the correlation coefficient decreases. A similar situation can be observed at the street canyon site (Fig 9 c, Table S2). This shows that for a concentration of SA higher than $10^{6.5}$, NPF processes are the dominating source of sub-3 nm particles at both sites. The slope and the correlation coefficient stay roughly constant in the correlation between sub-3 nm particles concentration and SA.

**Detailed minor comments:**

**L44: why should the adverse health effects of nanoparticles change the number of deaths. Please rephrase and specify why you think that the modeled numbers are biased low, and how this ties to your study.**

Thanks for the comment. We agree that excluding nanoparticles from models does not imply that models underestimate the number of deaths, because the adverse health impacts of nanoparticles are possibly included in the exposure-response functions (ERFs) of PM impacts on health: the concentrations of nanoparticles and particle mass presumably correlate between environments with different pollution levels. We rephrase this sentence:

Line 40: *Models show that outdoor air pollution causes approximately 400 000 premature deaths in Europe annually (Geels et al., 2015; Im et al., 2018), from which around 2000 occur in Finland (Im et al., 2019). However, nanoparticles have different health impacts than larger particles. Magnetite nanoparticles in urban air pollution accumulate in the brain and may cause neurodegenerative diseases (Maher et al., 2016). Nanoparticles with diameters below 3 nm may have significant, so far poorly understood, health effects due to their nose-to-brain transport via the olfactory pathway (Tian et al., 2019). Hence, a better understanding of the concentrations, sources and health impacts of the nanoparticles will improve the estimates of adverse health effects of urban pollution sources.*

**L54: I suggest to reorganize wording, put „emitted" , e.g.before "(Arnold)"**

We applied suggested change.

**L58 I suggest to remove "on the other hand". I don't see why the following statement is somewhat contra- the statement before L58**

We removed 'on the other hand' from this sentence.

**L64: it could be helpful to give an example for particle sources that are specific to traffic activities vs. traffic emissions, maybe even before this line. Readers might think this is interchangeable.**

Different sources of sub-3 nm particles are mentioned earlier in the paragraph, where we state: *The sub-3 nm particles are directly emitted from vehicle exhaust (Rönkkö et al., 2017; Sgro et al., 2012) and brake wear (Nosko et al., 2017) or formed in the exhaust plume from the nucleating gaseous components* (Rönkkö & Timonen, 2019). Hietikko et al. (2018) observed elevated concentrations of sub-3nm particles during rush hours and when the wind was coming from the direction of the road and concluded that this indicates that sub-3 nm particles are linked to traffic activities. To make this clear, we modified the sentence:

*Hietikko et al. (2018) observed elevated concentrations of sub-3 nm particles at the street canyon in Helsinki during rush hours and when wind was coming from the road, and concluded that sub-3 nm particles are linked to traffic activity. In addition, they found that high sub-3nm particle concentrations coincided with high concentrations of larger particles  suggesting the importance of traffic emissions*

**L67-68: this sounds over-generalized, and is not very helpful for the reader. The general implications for traffic and traffic related emissions on human health are not ambiguous, don't you agree?**

We agree that this sentence sounded over-generalized. We corrected it to the following form:

*Generally, the impacts of traffic on urban air quality and on human health is still not well quantified.*

**L76: I suggest to use plural for „distribution"**

We followed the reviewer's suggestion and changed the form to plural.

**L85: Wouldn't it be useful to now the measurement heights and also the heights and distances to the closest roads, for both measurement station, respectively?**

We agree that the distance from roads and measurement heights are useful parameters to know for the readers. That is why we added the distance between the street canyon site and the Mäkelänkatu road in the discussed line:

*The first one is the Helsinki Region Environmental Services (HSY) air quality station, which is placed in a street canyon (approximately 0.5 m from the edge of Mäkelänkatu street, and represents a busy street environment (with approximately 28 100 vehicles per workday) (Kuuluvainen et al., 2018).*

The distance and height between the background station and Hämeentie was already mentioned in lines 94-96:

*The SMEAR III is located on a hill, approximately 12 m above the nearest busy road (Hämeentie street). The SMEAR III is separated from Hämeentie by a 150 m band of a deciduous forest.*

We also included a sentence on the measurement heights to the *2.3. Particle size distribution measurement* section:

Line 119: *At the street canyon site, particle size distribution was measured at 4 m height, while at the background station at 1.5 m.*

**L150, correction described in limited detail. I suggest to describe more details of the procedure and the amount and variability of the resulting correction (could go in SI. At minimum the reader should know why this correction is important, how it changes the measurments and how certain you are about the correction result.**

We added a section in the supplementary information that describes this correction in details:

***Correction of UCPC data at the background station***

*During 29.05-06.04.2018 the nighttime ratio between PSM and UCPC concentrations as well as between UCPC and CPC concentrations drifted in opposite directions. At the same time, the ratio between PSM and CPC stayed constant, which indicated that the concentrations measured by UCPC drifted. This drift could be caused by a change in measuring parameter, for example, an increase in the aerosol flow. To correct this error, we divided the average nighttime ratio between UCPC and CPC from period 3.05-28.05.2018 by the nighttime ratio between these instruments for period 29.05-4.06.2018. We obtained the following correction parameters: 0.97 (29.05-30.05), 0.89 (31.05-01.06), 0.87 (02.06 and 04.06), and 0.92 (03.06). The distribution of the correction is presented in Fig S1 (a). The correction is much smaller than the standard deviation of UCPC data for the corrected period ($\sigma= 11\ 515\ cm^{-3}$). Figure S1 (b,c) present the difference in concentration measured by UCPC and CPC ($N_{3-7}$) as well as PSM and UCPC ($N_{1-3}$) during this campaign. UCPC correction is smaller than these concentrations, but it strongly influences the sub-3 nm concentrations measured for this period. This correction was applied only to a fraction of data at the background station, and it doesn't influence any analysis done for overlapping data between stations.*

[Figure]

*Figure S1. The distribution of the correction applied to data measured by UCPC during period 29.05-06.04.2018 (a), and the difference between concentrations measured by UCPC and CPC (b) as well as PSM and UCPC (c) during this campaign. Box plots present 25th, 50th (median), and 75th percentile values. The whiskers extend to the most extreme data points not considered outliers, and the outliers are plotted individually using the '+' symbol*

**L199, please specify what type of measurement the 50% uncertainty number refers to: an instantaneous measurement? an hourly average concentration? Is this significant for the purpose of your study?**

The uncertainty number refers to the average SA concentration. We clarified that in the manuscript. We also added a short explanation of the effect of this uncertainty on our analysis:

*Uncertainties of the averaged absolute concentration measured by CI-APi-TOF are in the order of 50%, while the uncertainties of relative changes in the concentration are smaller than 10% (Ehn et al., 2014), thus comparing SA concentration values between stations is not as accurate as analyzing time series of SA and other variables.*

**L228, doesn't this „decreasing trend with size" also apply to the night period? (this might be obscured by log plot). Looking at Figure 2, It also appears that the <100 nm fraction has a much smaller relative contribution to total N as compared to the 3 other time periods (noon, afternoon, night). That seems surprising to me, given that the morning rush hour related to 3 nm N as you describe in the following lines. (e.g. in L279 you say that "the diurnal variation of nucleation mode particle concentration is similar to that of sub 3nm particles…"; or looking at fig 4a, the largest N is observed for 3-25 nm around 05 to 08 local time).**

We agree that Figure 2 looks confusing while comparing it to Fig 4a. The different sample size causes the inconsistency between these two figures. Figure 2 is based on all available data for each station, while Figure 4 is split for weekdays and weekends. At the street canyon site, the majority of data points were measured during a weekend or a holiday. That explains why the size distribution at the street canyon, presented in Figure 2, does not resemble the diurnal variation of nucleation mode particles on workdays. We noticed that presenting these figures with different sample sizes can be confusing for a reader. Thus we decided to change Figure 2 so that it includes separate panels for weekends and workdays. We also changed figure S3, which presents data from Fig 2 with a linear y-axis.

[Figure]

*Figure 2. Median particle size distribution (a,c) at the street canyon and (b,d) at the background station. The colors indicate different periods of the day: night (1:00-4:00 LT, black), morning (6:00-9:00 LT, blue), noon (10:00-13:00 LT, green), and afternoon (15:00-18:00, red). The top row presents size distributions measured during weekends (a,b) and the bottom one during workdays (c,d). The median size distribution was determined by DMPS (particles with sizes between 6-800 nm) marked with solid lines in the figure, UCPC and CPC (3-7 nm), and PSM and UCPC (1-3 nm) marked with dots.*

Additionally, we made some changes in the text describing Fig. 2:

*The shape of the size distributions for Aitken (25-100 nm) and accumulation mode (100-800 nm) particles is quite similar at the two sites in most of the cases, but the concentrations are higher at the street canyon, as discussed later in this section. What stands out is almost constant Aitken mode measured during the daytime on workdays at the street canyon site. At the street canyon site, the concentration of particles in the nucleation mode (3-25 nm) has a decreasing trend with an increase of particle size, except for the nighttime on workdays. At the background station, during the night and afternoon, the concentration of particles in nucleation mode increases with increasing particle diameter. During noon the nucleation mode has a peak above 10 nm, likely linked to NPF events.*

**Fig2 – interestingly at the street site, the <100 nm fraction is the smallest in the morning I suspect that the axis label should read nm, please correct if wrong.**

As explained in the previous comment, the street canyon site size distribution was driven mostly by data measured during weekends. Fig 2 shows that the sub-100 nm fraction is smallest in the morning during weekends at the street canyon site. During workdays, the sub-100 nm fraction is the highest.

We apologize for the typo in the axis label. We corrected the axis label to nm.

**Fig.3, It would be helpful to know the difference in traffic density (not necessarily in real time from the time of measurement, but maybe from previous studies) between the two sites to interpret the significance of differences between blue and red lines.**

Hämeentien silta is the closest automatic traffic-counting site to the background station. It is located at the bridge at Hämeentie Street within 185 m from the background station. In May 2018, the traffic was measured there for the period of 24-30.05. Fig R2 presents the mean diurnal variation of traffic density for weekdays and weekends at Hämeentien silta. An automatic traffic-counting site located at Mäkelänkatu Street, close to the street canyon site, was malfunctioning during May 2018. That is why we used a previous study by Hietikko et al. (2018) as a reference for traffic density at the street canyon site. Hietikko et al. (2018) presents the diurnal variation for traffic density measured at Mäkelänkatu street in May 2017(Hietikko et al., 2018)(Hietikko et al., 2018) (Hietikko et al., 2018). The difference in traffic density between these stations is small and it should not affect the differences between the sites in Fig 3.

[Figure]

Figure R2. Mean diurnal variation of traffic density for weekdays (purple) and weekends (blue) at Hämeentien silta.

**Given that your sample size is not very large, how much do these figures change (especially plot c) if you limit you comparison to periods where all instruments at all sites were running simultaneously (i.e. around 100 or so hours for both sites). In the current version you subtract median statistics from a sample group "background" that is much larger (in sample size) than the sample group "street". How sensitives are the described and discussed patterns to meteorology etc. which might be different between the two sample groups.**

Figure R3 presents the diurnal variation of sub-3 nm particles at both stations for the overlapping data. The diurnal variations (Fig. R3 (a) and (b)) differed slightly from the one presented in Fig 3. At both stations, the morning rush hour peaks during workdays are weaker on Fig R3 than on Fig3. The difference in sub-3 nm particles between the sites during rush hours on workdays (Fig R3 (c)) is quite similar to the one presented in Fig. 3 (c). However, the diurnal variation of the subtracted concentrations (Fig R3 (c)) differ at 12-13 during workdays and at 12-15 during weekends from the diurnal variation presented in Fig. 3 (c). This can be explained by the stronger impact of NPF on the particle number concentrations in the overlapping data. During the time with overlapping data (172 hours), there was 2 NPF events. The effect of NPF events is also linked to meteorological conditions, as NPF events typically occur on sunny clear-sky days. On the other hand, the similarity of the particle concentrations during rush hours on workdays between the two data sets shows that human activity, thus traffic emissions, are relatively constant on workdays.

[Figure]

Figure R3. The diurnal variation of sub-3 nm particle concentration during weekends (red) and workdays (blue) (a) at the street canyon and (b) at the background station, and (c) the difference between median sub-3 nm particles concentration at the street canyon (a) and at the background station (b). The median diurnal variation is shown as a solid line with markers; the 25th and 75th percentile ranges are presented as shaded areas. This figure was made using only overlapping sub-3 nm particle data from both stations.

We also corrected the discussion of subplot 3 (c) in the manuscript:

*The difference between the stations (Fig. 3c) shows that median sub-3 nm particle concentrations during the rush hours in the street canyon site are clearly higher than at the background station throughout, roughly by a factor of 5. However, the concentrations are slightly higher also during nighttime, which shows the influence of the continuous traffic emissions at the street canyon site. One should keep in mind, though, that data presented in Fig. 3 can be affected by the differences in measurement period between the two sites, especially if the fraction of data collected during NPF event is different*

**L350, here your discussion could improve by a more quantitative description of the background conditions and confounding parameters, rather than stating "traffic volumes are lower", or "could be a plume from e.g. a ship". Can't you use your data and other data from Helsinki to investigate if this is a feasible explanation (wind direction, time of day).**

We agree that this is an interesting event; however, investigating the reasons for the observation in detail is outside the scope of this manuscript. Thus, we decided to not follow up on it.

**L337: While I can follow your observation of more peaks in N <3 nm as compared to SA at the street site, it would be much more informative to describe and analyze this in a more quantitative way. This seems to be an important result of your study, it could be "fleshed out" by more statistics and significance testing. Is really the majority of N not originating from SA (L337)? It seems to me that the largest peaks in N are accompanied by largest peaks in SA and that the peaks in N without a counterpart in SA are rather minor peaks. It's hard to judge by only looking at figures (thus my suggestion above).**

We agree that a more quantitative analysis will make this point clearer. That is why we added the correlation analysis between sub-3 nm particles and SA for both case studies. The analysis shows that the correlation between the concentrations of sub-3 nm particles and SA is weaker at the street canyon than at the background site, suggesting that SA is not originating from the direct traffic emissions, while sub-3 nm particles are. We add plot S10, showing these correlations, to the supplementary materials and the following sentence before the abovementioned line:

*The correlation between sub-3 nm particles and SA is much weaker at the street canyon site than at the background station for both case studies (Fig. S10).*

[Figure]

*Figure S10. Correlation between the sub-3 nm particles and SA measured at the background station (a,c) and at the street canyon (b,d) for period 5 May 2018-6 May 2018 LT (a,b) and 8 May 2018 09:00-9 May 2018 15:00 LT (c,d) colored by the time of the day. The black line shows the 1:1 line.*

**L376, I suggest to rephrase "number of studied points" to sample size Table 3, did you transform WD values before correlation? If so please state how in a footnote. If not, this correlation might not be useful.**

We changed it and use cosines and sines of wind direction for these correlations.

**L389, do you mean that bins were chosen to achieve similar classification sizes? What are "enough data points" ? Maybe just use group NOx low (n = xxx), vs. group SA low (n = yyy), with bin cut offs of *NOx < …) to clarify this in a non-wordy fashion.**

We rephrased this sentence:

*The bins were chosen for fitting so that the bin limits at both stations, and the sample size in each bin are comparable, and the bin width is either 0.25 or 0.5. The parameters characterizing the chosen bins are presented in Table S2.*

**L391, not only is the slope of sub 3nm vs NOx much smaller at the background site, your plot 9d also shows a much more varying slope as a function of SA at the background vs. the street site. Do you agree? Could this be addressed in your discussions following L391. For example, for the yellow and red dot clouds, the fits might be similar in slope for street vs background.**

Thanks for this comment, but we think there is not enough variation in the NOx concentration to see the differences in the slope clearly. One thing to keep in mind is that plots with a correlation between the

logarithm of $NO_x$ and the logarithm of sub-3 nm particles (9 c,d) have different limits on the x-axis. That was not mentioned in the article but we added this information to the caption of figure 9. Therefore, even if it looks lik the slope is similar at the two stations at logarithm of SA concentration higher than 6.75, it is not necessarily the case.The slope for the correlation between $\log_{10}(NO_x)$ and $\log_{10}(N_{1-3})$ at $\log_{10}(SA)$ higher than 6.75 is 1.07 at the background station and 2.28 at the street canyon station (Table S1).

**L418, Couldn't you study this possible relationship running a cross correlation analysis on your data? This ties to your discussion in line 467 about the possible traffic source of SA precursors.**

In order to try separating these sources, we analyzed the cross-correlation (Eq. R1) for SA and sub-3 nm particles as well as SA and $NO_x$ at both stations.

$$\hat{R}_{xy,normalized}(n) = \frac{\hat{R}_{xy}(n)}{\sqrt{\hat{R}_{xx}(n)\hat{R}_{yy}(n)}} \tag{R1}$$

where $\hat{R}_{xx}$ and $\hat{R}_{yy}$ are the autocorrelation of variables x and y, respectively and $\hat{R}_{xy}$ is the cross correlation of variables x and y calculated from Eq. R2:

$$\hat{R}_{xy,}(n) = \begin{cases} \sum_{n=0}^{N-n-1} x_{i+n}y_i^*, & n \geq 0, \\ \hat{R}_{xy,}(-n) & n < 0. \end{cases} \tag{R2}$$

In each analyzed case (fig R4-R5), the difference between cross-correlation results for different times (approximately 2%) is meaningless. We think that this analysis cannot help us to estimate how much SA concentration is coming from traffic.

[Figure]

Figure R4. Results of normalized cross-correlation analysis for SA and sub-3 nm particles concentrations (a) as well as SA and $NO_x$ concentrations for the street canyon site.

[Figure]

Figure R5. Results of normalized cross-correlation analysis for SA and sub-3 nm particles concentrations (a) as well as SA and NO$_x$ concentrations for the background station.

**Fig10, I wonder if you would achieve a better fit for the background station, only using SA as a predictor.**

Figure 10 shows an example of data with the fitting only for four days, as mentioned in the figure caption. We agree that based on this figure it seems that using only SA as a predictor could be as good as or ever better than using SA and NOx. We calculated RMSLE to the full dataset used for this fitting based on the Eq. R1:

$$RMSLE = \sqrt{\frac{1}{n}\sum_{i=1}^{n}(\log_{10}(\hat{y}_i + 1) - \log_{10}(y_i + 1))^2} \qquad \text{(R3)}$$

where $\hat{y}_i$ is a predicted value and $y_i$ is a measured value for sub-3 nm particles concentration for a data point i.

The RMSLE for using only SA as a predictor at background station is 0.4879, while the RMSLE for SA and NOx as predictors is equal 0.3709. That indicate that using both NOx and SA as predictors gives better fit than using only SA.

**Reply to anonymous referee #2**

**The manuscript by M. Okuljar and coauthors presents size distribution data over the size range between 1 and 800 nm from two urban locations in Helsinki. One station is located near a busy road while at the other measurement station urban background air is sampled. For the present study a multitude of instruments has been utilized to characterize particle size distribution, particle number concentration, selected gas phase compounds and meteorological parameters. The obtained particle data are categorized into different size ranges (sub-3 nm, nucleation mode, Aitken mode and accumulation mode) aiming at source apportionment between traffic related emissions and new particle formation (NPF). Apparently, traffic is a substantial local source of very small nanoparticles affecting sub-3 nm concentrations, nucleation mode and partly Aitken mode whereas accumulation mode particles are more related to long-range transport. Based on a statistical analysis the authors present a method by which the sub-3 nm concentration can be attributed to traffic and/or NPF. I think the manuscript clearly fits the scope of ACP and is of sufficient quality that allows publication with some minor corrections.**

**My only concern is about the dominant focus on the sub-3 nm particles. While the title emphasizes the full size range between 1 and 800nm, most of the analysis (>50%) only addresses the very smallest particles. I'd have preferred to see also some more information on the nucleation mode as this is also**

**clearly affected by traffic, maybe even more than the sub-3 nm particles (comparison of Figs. 3c and 4g). It would be interesting to see the regression analysis (section 3.5) also for nucleation mode and Aitken mode particles. If this is not intended for the current manuscript it should somehow reflect in the title (more focus on sub-3 nm).**

We agree that section 3.5 could also provide information about regression analysis for nucleation and Aitken modes. We added to Table 3 the results of regression analysis for particles with a diameter between 3 and 7 nm as well as for particles with a diameter in the range 7-25 nm. We chose these size ranges due to the differences in their behavior.

[revised manuscript text omitted]

**One other point relates to Figure 2. What causes the jump between the dots and lines? Could this be related to the conversion of CPC concentrations to dN/dlogDp? I guess there would also be some scatter on these data points that would maybe relax this sharp transition there.**

The difference is caused by uncertainties of sub-10 nm measurements with different instruments. This explanation was unclear in the manuscript, thus we rephrase it:

*A sudden change in concentrations of particles with a diameter below and above 7 nm at the background station can be associated with the uncertainty of different instruments used for measuring particles smaller than 10 nm (Kangasluoma et al., 2020).*

Related to this figure, see also the answers to Referee 1, to the question related to Line 228.

**Editorial suggestions:**

**Page 4, line 135: …particles of lower mobility… I'd replace "lower" by "corresponding"**

We replaced it as the reviewer suggested.

**Page 8, line 228: …particles in the nucleation mode…**

We added article 'the' as reviewer suggested.

**Page 8, line 233: …likely linked to NPF. ("an NPF event" does not make sense with the statistical analysis)**

It is true that the difference between NPF and NPF event was not clearly introduced in this manuscript, thus 'an NPF event' did not make sense with this analysis. We added the NPF event definition to the text (line 50):

*Regional NPF events, where the growth of the particles to larger sizes can be followed, are favored in specific meteorological conditions, for example, high solar radiation and low relative humidity, as well as an abundance of low-volatile gaseous precursors (Hussein et al., 2008; Kerminen et al., 2018; Salma et al., 2011; Wonaschütz et al., 2015).*

We also corrected the sentences in which 'NPF was misused. We replaced 'NPF' with NPF event' in following lines:

Line 57: *In many locations, SA concentration is one of the critical factors determining whether an NPF event occurs (Kuang et al., 2008; Ripamonti et al., 2013; Wang et al., 2011).*

Line 78: *In the Helsinki area, the focus of the research has been either on NPF events (Hussein et al., 2008, 2009) or the primary particle emissions (Hietikko et al., 2018; Ripamonti et al., 2013; Rönkkö et al., 2017).*

We also replaced 'NPF event' with 'NPF' in following sentence:

Line 57: *In addition to NPF, traffic is a significant source of the sub-3 nm particles (Hietikko et al., 2018; Rönkkö et al., 2017).*

**Page 8, line 235: …instruments particles… needs a preposition in between**

We agree that this sentence was unclear and confusing and we apologize for that. We clarified that sentence:

*A sudden change in concentrations of particles with a diameter below and above 7 nm at the background station can be associated with the uncertainty of the different instruments used for measuring particles smaller than 10 nm (Kangasluoma et al., 2020).*

**Page 11, line 319: …an order of magnitude higher… When looking at Figure 6 I would rather estimate a factor of 5 difference for night time sulfuric acid between street canyon and background.**

We agree that 'an order of magnitude higher' was an overstatement. We compared the data and SA median concentrations at street canyon site were maximum 3 times higher than at background station during nighttime. We changed this sentence:

*Nighttime median SA concentrations are as high at the street canyon site as at the background station.*

**Page 14, Table 3: The two asterisks next to SO2 are not explained.**

We added an explanation for the two asterisks.